# FedTreeLoRA: Reconciling Statistical and Functional Heterogeneity in Federated LoRA Fine-Tuning

**Jieming Bian** [* 1] **Lei Wang** [* 1] **Letian Zhang** [2] **Jie Xu** [1]

## Abstract

Federated Learning (FL) with Low-Rank Adaptation (LoRA) has become a standard for privacy-preserving LLM fine-tuning. However, existing personalized methods predominantly operated under a restrictive Flat-Model Assumption: they addressed client-side *statistical heterogeneity* but treated the model as a monolithic block, ignoring the *functional heterogeneity* across LLM layers. We argue that these two statistical (horizontal) and functional (vertical) dimensions, are *orthogonal in source yet coupled in interaction*, implying that the optimal depth of parameter sharing is functionally dependent on client similarity. To address this, we propose **FedTreeLoRA**, a framework employing tree-structured aggregation for fine-grained, layer-wise alignment. By dynamically constructing an aggregation hierarchy, FedTreeLoRA allows clients to share broad consensus on shallow 'trunks' while progressively specializing on deep 'branches'. Experiments on NLU and NLG benchmarks demonstrate that FedTreeLoRA significantly outperforms state-of-the-art methods by effectively reconciling generalization and personalization.

## 1. Introduction

Large Language Models (LLMs) have demonstrated significant potential across a wide range of domains, ranging from natural language understanding and code generation to complex mathematical reasoning (Devlin et al., 2019; Touvron et al., 2023a; Achiam et al., 2023; Touvron et al., 2023b; Team et al., 2023). To leverage these capabilities for specialized downstream applications, fine-tuning is often essential. However, the substantial scale of modern LLMs renders full fine-tuning computationally expensive and resource-intensive. Furthermore, in many real-world scenarios, high-quality domain data is dispersed across decentralized institutions or edge devices, where data sharing is restricted by privacy regulations and competitive concerns. Federated Learning (FL) (McMahan et al., 2017) has emerged as an effective paradigm to address these challenges, enabling collaborative training across distributed data sources without exchanging raw data. To make FL feasible for LLMs, Parameter-Efficient Fine-Tuning (PEFT) methods (Han et al., 2024), particularly Low-Rank Adaptation (LoRA) (Hu et al., 2021), are widely adopted to mitigate communication and computation overhead by updating only low-rank matrices while keeping pre-trained weights frozen.

Despite these advancements, applying LoRA in FL settings presents a critical challenge: *heterogeneity*. Existing works (Bian et al., 2025a) have predominantly focused on *horizontal heterogeneity*, which refers to the non-IID (non-independent and identically distributed) nature of client data. Early approaches (Zhang et al., 2024; Bian et al., 2025b; Wang et al., 2024b) aggregated all clients into a single global model, which often yields suboptimal performance on diverse data distributions. More recent personalized methods have attempted to mitigate this issue through distinct aggregation strategies. One line of research, represented by methods (Qi et al., 2024; Yang et al., 2024; Hao et al., 2025; Bian et al., 2026), employs a dual-module approach, maintaining separate global and local LoRA modules to balance generic knowledge sharing with personalized adaptation. Another line of research, exemplified by FedLEASE (Wang et al., 2025), addresses data heterogeneity by adaptively clustering clients based on representation similarity to train shared experts within groups. While these strategies improve upon simple global aggregation, they still rely on a restrictive **Flat-Model Assumption**: regardless of using dual modules or client clustering, they treat LoRA as a single monolithic unit, assuming that sharing decisions must remain uniform across layers.

We argue that this assumption overlooks a second, equally important dimension: the *Intra-Model Functional Heterogeneity* (which we term *vertical heterogeneity* in the context

*Equal contribution [1] University of Florida, Gainesville, FL 32611 [2] Middle Tennessee State University Murfreesboro, TN 37132. Correspondence to: Jieming Bian <jieming.bian@ufl.edu>, Lei Wang <leiwang1@ufl.edu>.

*Proceedings of the 43rd International Conference on Machine Learning*, Seoul, South Korea. PMLR 306, 2026. Copyright 2026 by the author(s).

of model depth). This is an architectural property, distinct from traditional vertical federated learning (data feature partitioning). Empirical studies indicate that LLM layers exhibit distinct representational properties: shallow layers typically extract general linguistic features, whereas deep layers encode specific semantic information (Gao et al., 2024; Qian et al., 2025). Consequently, the optimal degree of parameter sharing is not uniform but varies with model depth. By ignoring this vertical dimension, existing aggregation strategies face a fundamental dilemma: they either aggressively aggregate deep layers, risking negative transfer due to data conflicts, or conservatively isolate shallow layers, thereby under-utilizing shared general knowledge.

In this paper, we propose that these two dimensions of heterogeneity, the statistical *horizontal* and the functional *vertical*, are *orthogonal in source yet coupled in interaction*. They are orthogonal because horizontal heterogeneity stems from external data distributions, while vertical heterogeneity arises solely from the internal LLM architecture. However, they are coupled because the effective aggregation strategy depends on their intersection: the depth at which clients should cease sharing parameters is functionally dependent on their data similarity. To empirically verify this hypothesis, we first conduct extensive motivational studies analyzing the interaction between client data distributions and layer-wise representations. Our analysis confirms that clients with heterogeneous data distributions can benefit from sharing shallow 'trunks' of the model while diverging at deeper layers to preserve personalization. This implies that the optimal aggregation structure is not a set of disjoint clusters, but a hierarchical tree.

Based on these insights and empirical verifications, we introduce **FedTreeLoRA**, a novel framework that addresses this dual-heterogeneity through tree-structured aggregation. Unlike prior methods, including coarse-grained client grouping approaches such as FedLEASE (Wang et al., 2025), FedTreeLoRA implements a fine-grained, layer-wise alignment strategy. It utilizes a data-driven hierarchical approach to dynamically construct an aggregation tree, where the root represents shared shallow layers among all clients, and branches represent progressive specialization towards deeper layers for specific client subgroups. This structure naturally determines the optimal sharing boundary for each client group, ensuring that general capabilities are consolidated while specific data patterns are adapted locally. Our main contributions are summarized as follows:

- We identify the problem of *dual-heterogeneity* in federated LLM fine-tuning, revealing that ignoring the coupling between client-side data distribution and layer-wise representation leads to suboptimal aggregation.

- We propose **FedTreeLoRA**, a tree-structured frame-

work that adaptively determines the optimal parameter-sharing depth across layers, effectively reconciling generalization and personalization.

- We conduct extensive experiments on diverse benchmarks, demonstrating that FedTreeLoRA significantly outperforms state-of-the-art federated LoRA methods.

## 2. Related Works

**Personalized Federated Learning.** Federated Learning (McMahan et al., 2017; Huang et al., 2026; Wang et al., 2024a; Bian et al., 2024; Zhang et al., 2025a; Liu et al., 2024; 2025; Peng et al., 2024) enables collaborative training while preserving data privacy, yet it faces significant challenges from statistical data heterogeneity (Li et al., 2020). To address this, Personalized FL (PFL) has been extensively studied, employing techniques such as regularization (Karimireddy et al., 2020), meta-learning (Fallah et al., 2020), and clustering-based methods (Ghosh et al., 2020; Sattler et al., 2020) that group clients with similar distributions. While prior work has explored *layer-wise* aggregation strategies, selectively sharing shallow layers while personalizing deep ones, these have been almost exclusively limited to small models like CNNs (e.g., FedPer (Arivazhagan et al., 2019), LG-FedAvg (Liang et al., 2020)). However, extending these insights to LLMs is non-trivial. Unlike CNNs, where hierarchy is defined by spatial feature progression, Transformer-based LLMs consist of architecturally identical layers where hierarchy emerges from *semantic specialization* (from syntax to reasoning) (Jawahar et al., 2019; Gao et al., 2024). To the best of our knowledge, no prior work has systematically examined how LoRA parameters should be aggregated differentially across Transformer depth based on client similarity, a gap this work aims to fill.

**Federated Fine-tuning with LoRA.** Given the immense computational cost of full LLM fine-tuning, integrating Parameter-Efficient Fine-Tuning methods (Houlsby et al., 2019), particularly Low-Rank Adaptation (Hu et al., 2021), into FL has become a dominant paradigm. Early approaches, such as FedIT (Zhang et al., 2024) and SLoRA (Babakniya et al., 2023), focused on training a single shared global LoRA module. However, these global methods often struggle with domain shifts in real-world heterogeneous scenarios. Consequently, recent research has pivoted towards personalized federated LoRA. FedSA (Guo et al., 2025), proposes splitting the LoRA module (e.g., aggregating matrix A while keeping B local). Another direction, exemplified by FedDPA (Yang et al., 2024) and FedALT (Bian et al., 2026), utilizes a dual-branch architecture containing both global and local LoRA modules. More recently, clustering-based methods like FedLEASE (Wang et al., 2025) group clients to train shared domain experts. Despite their architectural differences, these personalized methods share a common

limitation: they rely on a **Flat-Model Assumption**. They treat the LoRA module as a monolithic unit, overlooking the intrinsic *vertical heterogeneity* of LLMs. Our proposed **FedTreeLoRA** challenges this assumption by introducing a tree-structured aggregation mechanism that explicitly aligns aggregation depth with client similarity.

## 3. Preliminaries

### 3.1. Low-Rank Adaptation (LoRA) for LLMs

LoRA (Hu et al., 2021) is a parameter-efficient fine-tuning method predicated on the hypothesis that the change in weights during model adaptation has a low intrinsic dimension. For a pre-trained Transformer model consisting of $L$ layers, let $W_0 \in \mathbb{R}^{d_{out} \times d_{in}}$ denote a specific frozen weight matrix (e.g., a query or value projection) within a given layer. LoRA bypasses the update of $W_0$ by injecting two trainable low-rank decomposition matrices, $B \in \mathbb{R}^{d_{out} \times r}$ and $A \in \mathbb{R}^{r \times d_{in}}$, where the rank $r \ll \min(d_{in}, d_{out})$.

The forward pass for an input $x \in \mathbb{R}^{d_{in}}$ is modified as:

$$h = W_0 x + \Delta W x = W_0 x + B A x, \qquad (1)$$

where $A$ is typically initialized with a random Gaussian distribution and $B$ is initialized to zero, ensuring that the training begins with the original model behavior (i.e., $\Delta W = 0$). We denote the collection of all LoRA adapters across the model layers as $\Theta$.

### 3.2. Problem Formulation: Federated Fine-tuning

We consider a federated learning system comprising $N$ clients, indexed by $k \in \{1, \ldots, N\}$, each possessing a private dataset $\mathcal{D}_k$ drawn from a heterogeneous distribution $\mathcal{P}_k$. In the context of personalized federated fine-tuning, each client $k$ aims to obtain a specific set of effective LoRA parameters, denoted as $\Theta_k$, to adapt the shared frozen backbone $\mathcal{W}_0$. The overarching goal is to solve the objective:

$$\min_{\{\Theta_k\}_{k=1}^{N}} \sum_{k=1}^{N} p_k \mathbb{E}_{\xi \sim \mathcal{D}_k} [\ell(\Theta_k; \mathcal{W}_0, \xi)], \qquad (2)$$

where $p_k$ is the relative weight of client $k$. Crucially, in our framework, $\Theta_k$ is not necessarily trained in isolation. Instead, it is a composite mapping derived from the federated aggregation mechanism. Our specific goal is to design a structural dependence such that $\Theta_k$ leverages shared knowledge from similar clients (via the aggregation tree) while retaining task-specific adaptability.

## 4. Motivational Studies

While the hierarchical nature of LLMs is well-established in centralized settings (Jawahar et al., 2019), its implications for federated adaptation remain under-investigated.

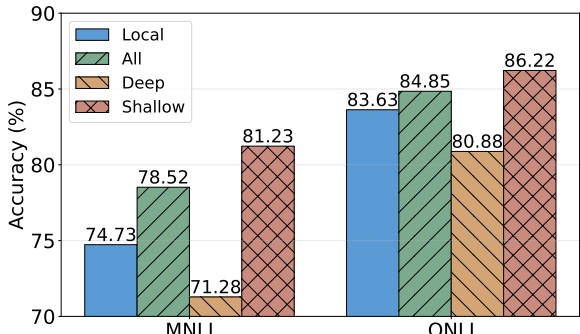

*Figure 1.* **Vertical Heterogeneity.** Aggregating only shallow layers significantly outperforms aggregating deep layers.

We posit that treating all LoRA layers uniformly, as done in current **Flat-Model** approaches, leads to inefficient parameter sharing. In this section, we conduct two motivational studies to quantify how *vertical heterogeneity* (layer depth) interacts with *horizontal heterogeneity* (client data distribution) to determine the optimal aggregation strategy. Due to space constraints, we provide the full experimental setup and extended results in Appendix B.

**Observation 1: Vertical (Intra-Model Functional) heterogeneity dictates the stability of federated aggregation.** We fine-tune a RoBERTa-Large (Liu et al., 2019) model on the GLUE benchmark (Wang et al., 2018) with ten clients partitioned under a non-IID distribution (Dirichlet $\alpha = 0.5$). We vary the aggregation scope: aggregating only shallow layers, aggregating only deep layers, full model aggregation, and purely local training. As illustrated in Figure 1, aggregating shallow layers yields significantly higher accuracy than aggregating deep layers. Crucially, aggregating deep layers is *detrimental*, resulting in performance even worse than local training. Furthermore, full model aggregation under heterogeneous conditions degrades performance relative to the shallow-only strategy. These results indicate that the hierarchical nature of LLMs directly translates into aggregation sensitivity: deep layers are highly vulnerable to negative transfer when clients diverge, whereas shallow layers provide a robust common ground for collaboration. Thus, *vertical heterogeneity* is not merely an architectural feature but a critical determinant of effective parameter sharing and aggregation stability.

**Observation 2: The optimal sharing boundary is coupled with horizontal heterogeneity.** We further investigate the interaction between vertical depth and statistical divergence through a controlled study. Specifically, we evaluate two clients on the MNLI dataset using RoBERTa-Large across three distinct regimes: *Homogeneous*, *Moderate*, and *Heterogeneous*. We vary the aggregation scope to encompass the first 8, 16, or all 24 layers. As illustrated in Figure 2, we observe a distinct shift in the optimal aggregation depth

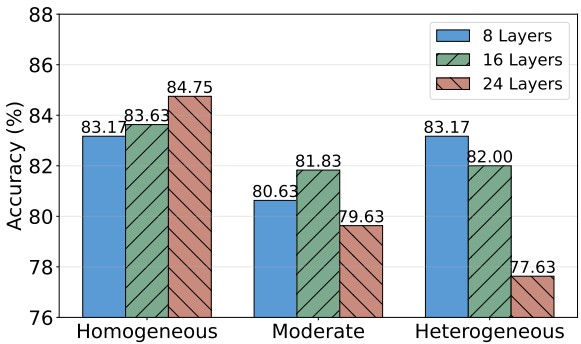

*Figure 2.* **The Coupling Effect of Dual Heterogeneity.** As client distributions diverge (from Homogeneous to Heterogeneous), the optimal sharing boundary shifts from deep to shallow layers.

based on the severity of data heterogeneity. In the *Homogeneous* setting, performance improves monotonically as aggregation extends to deeper layers, benefiting from maximum knowledge sharing. However, as heterogeneity increases, the performance peak shifts sharply toward shallower layers. In the *Heterogeneous* setting, aggregating beyond the shallow layers becomes detrimental, yielding results inferior to those of partial aggregation. These findings demonstrate that the "safe" depth for parameter sharing is not an architectural constant but is dynamically coupled with the statistical similarity between clients.

This coupling implies that a valid aggregation strategy must determine a hierarchical sharing boundary rather than simply grouping clients into disjoint, flat clusters as in prior work (Wang et al., 2025). This motivates the design of **FedTreeLoRA**, which constructs a tree-structured mechanism to ensure that general capabilities are shared at the root while task-specific reasoning diverges at the branches.

# 5. Methodology: FedTreeLoRA

We introduce FedTreeLoRA to reconcile statistical and functional heterogeneity. Unlike static flat-model approaches, FedTreeLoRA leverages a global dependency tree to enforce structural consistency while enabling adaptive, layer-wise granularity control across the Transformer architecture.

## 5.1. Global Topological Structure Modeling

The foundation of our approach is the construction of a global hierarchy that captures the overall statistical relationships among clients. The process begins with a warmup phase where each client $k \in \{1, \ldots, N\}$ independently performs local fine-tuning on the pre-trained backbone using its private dataset $\mathcal{D}_k$ for $E_{warm}$ epochs. This yields an initial set of LoRA parameters. Consistent with prior findings that the $B$ matrices capture task-specific semantic variations (Tian et al., 2024), we utilize the accumulated updates in $B$

to quantify client similarity, and we further provide a layer-wise heterogeneity analysis in Appendix G to empirically justify this choice. Specifically, we compute the *Global Distance Matrix* $D^{global} \in \mathbb{R}^{N \times N}$ via the average distance between client updates:

$$D_{i,j}^{global} = \frac{1}{L} \sum_{l=1}^{L} \text{dist}(B_{l,i}, B_{l,j}), \quad (3)$$

where $L$ denotes the total number of layers and $\text{dist}(\cdot, \cdot)$ is a distance operator in the parameter space. We default to the Frobenius distance due to its robustness in high-dimensional spaces (Aggarwal et al., 2001). Alternatives like cosine distance are evaluated in Section C.4.

We construct a binary merge tree $\mathcal{T}$ via Agglomerative Hierarchical Clustering (AHC) on $D^{global}$, encoding a spectrum of partitions from $P_1$ (universal sharing) to $P_N$ (full personalization). Crucially, constructing $\mathcal{T}$ from global rather than layer-wise information prevents *topological incoherence*, where contradictory client groupings across adjacent layers disrupt semantic continuity (e.g., clusters $\{1, 2\}$ and $\{3, 4\}$ at layer $l$ are reshuffled into $\{1, 3\}$ and $\{2, 4\}$ at layer $l + 1$). By establishing this global skeleton, we ensure every layer-specific clustering is a valid cut of a unified topology, guaranteeing structural consistency: clients separated at shallow layers remain specialized at deeper layers, thereby preserving the logical hierarchy of expert specialization.

## 5.2. Adaptive Layer-wise Depth Alignment

While the global tree $\mathcal{T}$ provides structural candidates, the optimal aggregation resolution varies by depth, shifting from broad consensus in shallow layers to fine-grained specialization in deep ones. To identify the optimal cluster count $c_l^*$ for layer $l$, we first quantify local functional heterogeneity via the *Layer-wise Distance Matrix* $D^{(l)}$:

$$D_{i,j}^{(l)} = \text{dist}(B_{l,i}, B_{l,j}). \quad (4)$$

Unlike the global metric, $D^{(l)}$ captures layer-specific data distributions, serving as the basis for evaluating the fitness of any candidate partition $P_c$ at this specific depth.

To enforce the hierarchical prior that specialization increases with depth, we impose a monotonicity constraint: $c_l \geq c_{l-1}$. We set $c_0^* = 1$ to represent the shared global root. For each Transformer layer $l \in \{1, \ldots, L\}$, given the optimal $c_{l-1}^*$ from the previous layer, we define the feasible *search space* $\Omega_l$, constrained by the window size $K$:

$$\Omega_l = \{c \in \mathbb{Z} \mid c_{l-1}^* \leq c < \min(N, c_{l-1}^* + K)\}. \quad (5)$$

This formulation ensures that even for the first layer ($l = 1$), the algorithm dynamically searches for the optimal granularity starting from $c = 1$ (global sharing) up to $K$, allowing

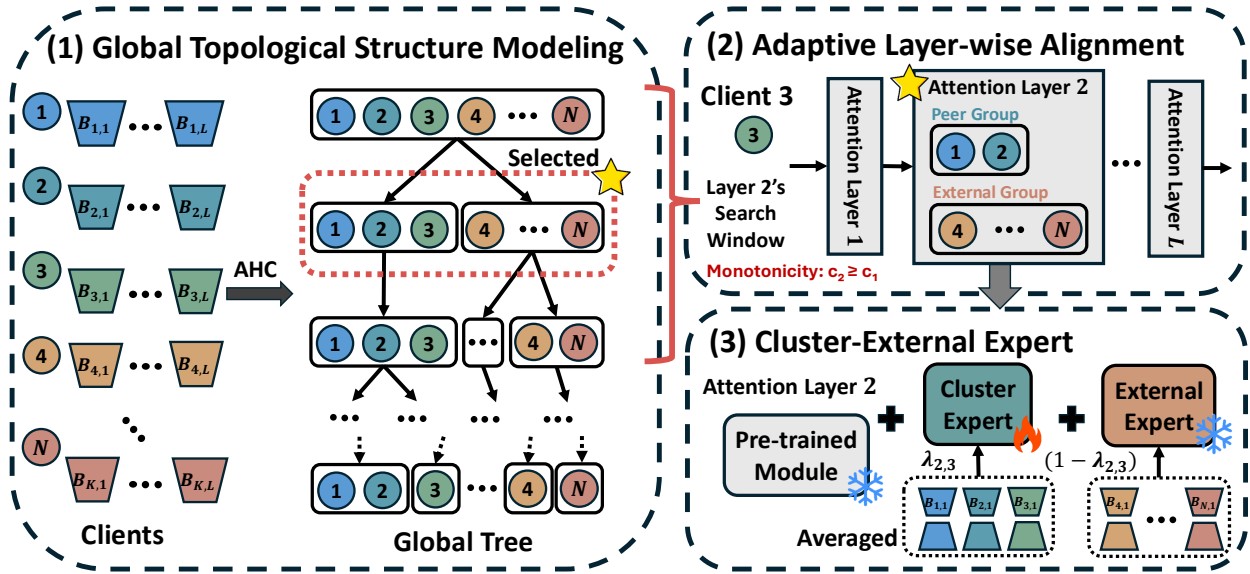

*Figure 3.* Overview of FedTreeLoRA. **(1) Global Topological Structure Modeling**: A hierarchy tree is built via AHC on client LoRA $B$ matrices during warmup to capture cross-client relationships. **(2) Adaptive Layer-wise Alignment**: For each layer $l$, the optimal cluster count $c_l^*$ is dynamically selected under a monotonicity constraint. **(3) Cluster-External Expert Mechanism**: Each client synthesizes parameters by mixing a Cluster Expert with an External Expert via a learnable coefficient $\lambda_{l,k}$.

the model to maintain the current granularity or refine it by splitting existing clusters.

We select the optimal cluster count from $\Omega_l$ using the Silhouette Coefficient (Rousseeuw, 1987), denoted as $\mathrm{Sil}(P_c, D^{(l)})$, which measures cluster validity based on the layer-specific distance. To handle the singleton case ($c = 1$) where the Silhouette score is undefined, we introduce a heterogeneity threshold $\tau$ as a baseline. We define the scoring function $\phi(c; D^{(l)})$ as $\tau$ if $c = 1$, and $\mathrm{Sil}(P_c, D^{(l)})$ otherwise. This threshold $\tau$ effectively controls the resistance to splitting; the algorithm transitions from a global shared model to specialized clusters only when the distinguishability of client distributions exceeds $\tau$. The optimal cluster count for layer $l$ is thus determined by:

$$c_l^* = \underset{c \in \Omega_l}{\arg\max} \, \phi(c; D^{(l)}). \tag{6}$$

This sequential optimization ensures that the aggregation structure evolves dynamically from the root to the leaves, strictly governed by the functional properties of each layer.

### 5.3. Cluster-External Expert Mechanism

The layer-wise partitions derived in Sec. 5.2 provide a topological blueprint of client relationships, specifically identifying peer groups with high functional similarity. To operationalize this topology for federated fine-tuning, we must define how parameters are synthesized based on these structural priors. While our tree-structured alignment is compatible with various interaction mechanisms (as discussed later),

we propose a parameter-efficient *Cluster-External* aggregation strategy as the primary instantiation. This design allows clients to leverage high-fidelity consensus from their specific peer cluster while retaining a pathway to broad global knowledge, effectively preventing information isolation without incurring the high computational cost of complex routing.

For a given layer $l$, the optimal cluster count $c_l^*$ induces the partition $P_{c_l^*} = \{\mathcal{C}_1, \ldots, \mathcal{C}_{c_l^*}\}$. For any client $k$, let $\mathcal{S}_k^{(l)} \in P_{c_l^*}$ denote the cluster containing $k$ (peer group), and let $\mathcal{R}_k^{(l)}$ denote the remaining clients (external group).

**Expert Construction.** Instead of aggregating generic parameter sets, we explicitly construct two specific LoRA experts for client $k$ at layer $l$: the **Cluster Expert** and the **External Expert**. Let $\Phi \in \{A, B\}$ denote a LoRA parameter matrix. We compute the aggregated experts as:

$$
\begin{aligned}
\bar{\Phi}_{l,k}^{\mathrm{clus}} &= \frac{1}{|\mathcal{S}_k^{(l)}|} \sum_{j \in \mathcal{S}_k^{(l)}} \Phi_{l,j}, \\
\bar{\Phi}_{l,k}^{\mathrm{ext}} &= \frac{1}{|\mathcal{R}_k^{(l)}|} \sum_{j \in \mathcal{R}_k^{(l)}} \Phi_{l,j}.
\end{aligned}
\tag{7}
$$

Note that if $\mathcal{S}_k^{(l)}$ contains all clients (e.g., at the root layer), the External Expert is zeroed to avoid redundancy.

**Forward Process and Updating.** To balance peer-group specialization and global knowledge sharing, we introduce a learnable mixing coefficient $\lambda_{l,k} \in [0, 1]$ for each layer. The forward pass for input $x$ at layer $l$ combines the frozen

pre-trained weight $W_{0,l}$ with the weighted contributions of the Cluster and External Experts:

$$h_l(x) = W_{0,l}x + \lambda_{l,k}\left(\bar{B}_{l,k}^{\text{clus}}\bar{A}_{l,k}^{\text{clus}}x\right) \\ + (1 - \lambda_{l,k})\left(\bar{B}_{l,k}^{\text{ext}}\bar{A}_{l,k}^{\text{ext}}x\right). \quad (8)$$

During local training, client $k$ updates **only** the Cluster Expert parameters $(\bar{A}_{l,k}^{\text{clus}}, \bar{B}_{l,k}^{\text{clus}})$ and the coefficient $\lambda_{l,k}$, while keeping the External Expert frozen. This ensures that the client refines the consensus of its peer group while retaining static access to broader global features.

Crucially, the proposed tree structure defines the *topology of sharing*, specifically identifying optimal peer groups for constructing layer-wise Cluster Experts. This topological contribution is orthogonal to the specific *combination strategy* used to integrate these experts during inference. While Eq. (8) adopts a parameter-efficient scalar mixing approach with a consolidated External Expert to minimize communication and computational overhead, our framework is inherently compatible with diverse interaction mechanisms. Specifically, the framework can be instantiated with: (1) a learnable Mixture-of-Experts (MoE) router (Jordan & Jacobs, 1994) replacing the scalar coefficient $\lambda$ for input-dependent dynamic selection; (2) a decomposed set of distinct external experts corresponding to specific peer clusters, rather than a single aggregated External Expert; or (3) an isolationist strategy that utilizes solely the Cluster Expert, discarding external information entirely. In our experiments, we explore these variations to analyze the trade-offs between performance and efficiency, demonstrating that the benefits of our layer-wise topological alignment persist regardless of the specific combination operator employed.

### 5.4. Convergence Analysis

Our analysis focuses on the proposed adaptive update rule, where the effective weight update is driven by the trainable Cluster Expert while conditioned on the frozen External Expert. To facilitate the analysis, we adopt standard assumptions commonly used in federated optimization: local objective functions are $\sigma$-smooth and stochastic gradients are unbiased with bounded variance $G^2$ (Assumptions A.1 and A.2 in Appendix). In addition, specific to low-rank adaptation, we follow the formulation in (Guo et al., 2025) and assume that the LoRA matrices are bounded by constants $M_A$ and $M_B$, and satisfy a gradient-alignment condition with coefficients $\mu_A, \mu_B > 0$, ensuring that optimization within the low-rank subspace yields meaningful descent directions for the full parameter space (Assumption A.3).

**Theorem 5.1.** *Let Assumptions A.1–A.3 hold. Let E be the number of local SGD steps and choose stepsize $\eta > 0$. Define the composite constant $\Gamma$ collecting all $O(\eta^2)$ terms from local updates and tree-structured aggregation as:*

$$\Gamma = LM_A^2M_B^2G^2 + C\,\sigma LG^2\left(M_A^4 + M_B^4 + M_A^4M_B^4\right),$$

*for some constant $C > 0$. The average squared gradient norm of the iterates generated by FedTreeLoRA satisfies*

$$\frac{1}{NT}\sum_{k=1}^{N}\sum_{t=1}^{T}\mathbb{E}\left[\|\nabla\mathcal{L}_k(\mathbf{W}_k^{(t)})\|_F^2\right] \leq \frac{2}{\mu_A + \mu_B}\sqrt{\frac{\Delta\cdot\Gamma}{T}},$$

*where $\Delta$ denotes the initial optimality gap.*

Theorem 5.1 shows an $\mathcal{O}(1/\sqrt{T})$ convergence rate under standard smooth non-convex assumptions, matching FedAvg (Yu et al., 2019) and FedSA (Guo et al., 2025). Detailed assumptions and proofs are provided in Appendix A.

## 6. Experiments

**Baseline Methods.** We compare against representative baselines categorized into two groups: *General Federated Fine-Tuning:* (1) **FedIT** (Zhang et al., 2024): Applies FedAvg directly to LoRA parameters; (2) **FFA-LoRA** (Sun et al., 2024): Freezes $A$ matrices and fine-tunes only $B$ for communication efficiency. *Personalized Federated Fine-Tuning:* (3) **FedSA** (Guo et al., 2025): Aggregates global $A$ matrices while keeping $B$ local; (4) **FedDPA** (Yang et al., 2024): Decouples knowledge via dual global and local modules; (5) **FedALT** (Bian et al., 2026): Mixes continuously trained local modules with frozen Rest-of-World parameters; (6) **FedLEASE** (Wang et al., 2025): A strong flat-clustering baseline that trains shared domain experts, serving as a direct contrast to our hierarchical approach. Detailed descriptions of these baseline methods and their specific implementation settings are provided in Appendix H.

### 6.1. Natural Language Understanding

**NLU Setup.** We employ RoBERTa-Large (355M) (Liu et al., 2019), consisting of 24 Transformer layers, as the backbone model. Evaluation is conducted on four datasets from the GLUE benchmark (Wang et al., 2018): QQP, QNLI, MNLI, and SST2. We simulate a federated system with $N = 20$ clients, each possessing 1,000 training samples. To emulate statistical heterogeneity, data is partitioned using a Dirichlet distribution with $\alpha = 0.5$. Training is performed with a batch size of 128 for $E = 2$ local epochs over $T = 30$ communication rounds. LoRA adapters with rank $r = 4$ are applied to the query and value projections, while the classification head remains frozen. Learning rates are tuned via grid search over $\eta \in \{1\text{E}{-}4, 3\text{E}{-}4, 5\text{E}{-}4, 1\text{E}{-}3, 3\text{E}{-}3, 5\text{E}{-}3\}$ for each method. We report Accuracy as the evaluation metric. All reported results are averaged over three independent runs.

**Performance Comparison.** Table 1 presents the comparative results on the NLU benchmarks under the non-IID setting. Several key observations arise from this evaluation. First, personalized federated fine-tuning methods

*Table 1.* Performance comparison on GLUE benchmarks (RoBERTa-Large-355M) under non-IID settings ($\alpha = 0.5$).

| Methods | Rank | % Param | MNLI | QNLI | SST2 | QQP | Average | $\Delta$ |
|---|---|---|---|---|---|---|---|---|
| FedIT (Zhang et al., 2024) | 4 | 0.1107% | $83.18 \pm 0.74$ | $87.03 \pm 0.43$ | $93.65 \pm 0.63$ | $84.93 \pm 0.59$ | 87.20 | - |
| FFA-LoRA (Sun et al., 2024) | 4 | 0.0553% | $83.02 \pm 0.42$ | $87.47 \pm 0.13$ | $93.73 \pm 0.18$ | $83.48 \pm 0.36$ | 86.93 | $-0.27$ |
|  | 8 | 0.1107% | $83.13 \pm 0.56$ | $88.90 \pm 0.30$ | $93.95 \pm 0.20$ | $83.62 \pm 0.48$ | 87.40 | $+0.20$ |
| FedSA (Guo et al., 2025) | 4 | 0.1107% | $83.63 \pm 0.69$ | $91.32 \pm 0.10$ | $95.87 \pm 0.30$ | $89.33 \pm 0.43$ | 90.04 | $+2.84$ |
| FedDPA (Yang et al., 2024) | 4 | 0.1107% | $83.97 \pm 0.75$ | $91.31 \pm 0.51$ | $95.72 \pm 0.12$ | $89.74 \pm 0.58$ | 90.19 | $+2.99$ |
| FedALT (Bian et al., 2026) | 4 | 0.1383% | $84.03 \pm 0.59$ | $90.77 \pm 0.46$ | $96.16 \pm 0.29$ | $89.27 \pm 0.50$ | 90.06 | $+2.86$ |
| FedLEASE (Wang et al., 2025) | 4 | 0.1521% | $86.21 \pm 0.36$ | $92.56 \pm 0.77$ | $95.63 \pm 0.34$ | $90.36 \pm 0.46$ | 91.19 | $+3.99$ |
| **FedTreeLoRA (Ours)** | 4 | 0.1107% | $\mathbf{88.15 \pm 0.25}$ | $\mathbf{93.37 \pm 0.62}$ | $\mathbf{96.56 \pm 0.07}$ | $\mathbf{91.35 \pm 0.17}$ | **92.36** | **+5.16** |

*Table 2.* Performance comparison on FLAN benchmarks (LLaMA-2-7B). We report ROUGE-1 scores.

| Methods | Rank | % Param | Text Edit | Struct2Text | Sentiment | Reasoning | Average | $\Delta$ |
|---|---|---|---|---|---|---|---|---|
| FedIT (Zhang et al., 2024) | 8 | 0.0622% | $59.84 \pm 1.17$ | $51.71 \pm 1.14$ | $44.53 \pm 1.30$ | $74.42 \pm 0.74$ | 57.62 | - |
| FFA-LoRA (Sun et al., 2024) | 8 | 0.0311% | $58.64 \pm 0.96$ | $51.83 \pm 0.48$ | $44.26 \pm 0.31$ | $73.62 \pm 0.43$ | 57.09 | $-0.53$ |
|  | 16 | 0.0622% | $59.15 \pm 0.16$ | $52.85 \pm 0.54$ | $44.96 \pm 2.22$ | $73.77 \pm 0.28$ | 57.68 | $+0.06$ |
| FedSA (Guo et al., 2025) | 8 | 0.0622% | $63.80 \pm 1.88$ | $54.48 \pm 0.53$ | $46.98 \pm 0.55$ | $74.37 \pm 0.94$ | 59.91 | $+2.29$ |
| FedDPA (Yang et al., 2024) | 8 | 0.0622% | $64.33 \pm 0.92$ | $54.18 \pm 1.12$ | $48.13 \pm 1.02$ | $75.55 \pm 1.43$ | 60.55 | $+2.93$ |
| FedALT (Bian et al., 2026) | 8 | 0.0699% | $67.61 \pm 1.80$ | $54.06 \pm 1.51$ | $48.57 \pm 1.26$ | $76.84 \pm 1.48$ | 61.77 | $+4.15$ |
| FedLEASE (Wang et al., 2025) | 8 | 0.0895% | $66.31 \pm 1.43$ | $54.80 \pm 1.07$ | $49.32 \pm 0.88$ | $76.40 \pm 1.07$ | 61.71 | $+4.09$ |
| **FedTreeLoRA (Ours)** | 8 | 0.0622% | $\mathbf{68.63 \pm 0.87}$ | $\mathbf{55.59 \pm 0.86}$ | $\mathbf{51.27 \pm 0.71}$ | $\mathbf{77.27 \pm 0.73}$ | **63.19** | **+5.57** |

consistently outperform general approaches (e.g., FedIT), validating the necessity of personalization in heterogeneous environments. Second, regarding efficiency-performance trade-offs, FFA-LoRA yields suboptimal results; even when doubling the rank to $r = 8$ to equalize the trainable parameter count with other baselines, it fails to recover the performance drop caused by freezing the $A$ matrices.

Third, while FedLEASE outperforms prior baselines by leveraging clustering and adaptive expert selection, this comes at the cost of significantly increased training workload due to the additional parameters required for its adaptive top-M routers. In sharp contrast, FedTreeLoRA achieves state-of-the-art performance across all four tasks with negligible parameter overhead (limited to layer-wise scalar mixing coefficients $\lambda_{l,k}$). This result is theoretically significant: it demonstrates that the performance bottleneck in prior personalized methods stems from focusing solely on *horizontal* (statistical) heterogeneity. By explicitly addressing *vertical* (functional) heterogeneity via our tree-structured alignment, FedTreeLoRA unlocks superior generalization and personalization efficiency without the computational burden of complex routing networks.

**Visualization of Adaptive Structure.** To verify that FedTreeLoRA dynamically adapts to diverse data distributions, we visualize the layer-wise evolution of the cluster count ($c_l^*$) across the evaluated datasets in Figure 4. The distinct trajectories observed for each dataset confirm that our framework does not enforce a rigid template; instead, it automatically learns a tailored topological structure driven by the specific heterogeneity profile of each condition. We fur-

ther provide detailed structural visualizations under extreme task heterogeneity in Appendix C.

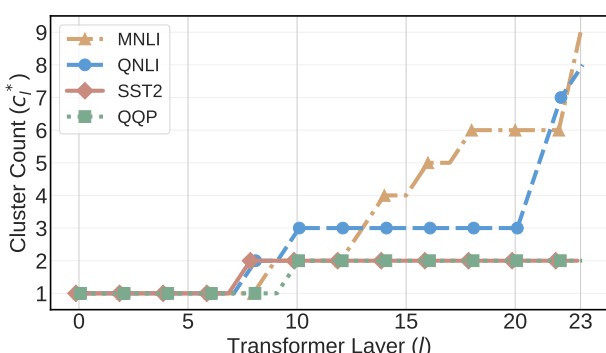

*Figure 4.* Layer-wise cluster counts ($c_l^*$) across different datasets. FedTreeLoRA adaptively identifies the optimal aggregation granularity specific to each data distribution.

**Ablation Studies.** We conduct comprehensive ablations on GLUE to isolate the contribution of each component in FedTreeLoRA. For brevity, we report average accuracy here, while detailed task-wise results are provided in Appendix D.

*(1) Impact of Interaction Mechanisms.* We compare the scalar-mixed *Cluster-External* aggregation with three alternatives enabled by our topological framework: **Isolationist** (Cluster Expert only), **Decomposed Experts** (distinct experts for every peer cluster), and **MoE Router** (replacing $\lambda_{l,k}$ with a learnable MLP router (Jordan & Jacobs, 1994)). As shown in Table 3, the Decomposed Experts variant achieves the highest average accuracy (92.57%), but incurs substantial communication overhead due to exchang-

ing multiple expert modules. Interestingly, the Isolationist variant still attains 91.40%, surpassing the strongest baseline (FedLEASE, 91.19%, Table 1). This highlights that our tree-structured alignment (Sec. 5.2) is the dominant performance contributor: precise peer matching already mitigates most heterogeneity without complex routing or dual branches. Finally, Scalar-Mixed achieves the best practicality–performance trade-off, recovering most of the remaining gap with *negligible* parameter overhead ($\approx 0.020\%$) and minimal communication cost, making it highly suitable for resource-constrained federated environments.

*Table 3.* Ablation on interaction mechanisms. "Params $\uparrow$" denotes the relative increase in trainable parameters vs. standard LoRA.

| Interaction Variant | Params $\uparrow$ | Comm. | Avg. Acc. |
|---|---|---|---|
| Isolationist (Cluster-Only) | 0% | Low | 91.40 |
| Decomposed Experts | $\approx 0.028\%$ | High | **92.57** |
| MoE Router | 25% | Low | 92.02 |
| **Scalar-Mixed (Ours)** | $\approx 0.020\%$ | Low | 92.36 |

*(2) Effect of Layer-wise Adaptivity.* A core premise of our work is that optimal aggregation granularity varies with model depth to address vertical heterogeneity. To validate this, we compare our adaptive depth search ($c_l^*$) against *Fixed-Depth* strategies, where we force a uniform number of clusters ($k = 1, 4, 8$) across all layers. As shown in Table 4, applying a uniform clustering depth yields suboptimal results. A global model ($k = 1$), such as FedIT, underfits deep-layer heterogeneity, while enforcing high granularity ($k = 8$ everywhere) harms shallow-layer representation learning due to data fragmentation. Our adaptive approach outperforms the best fixed setting, confirming that decoupling the aggregation depth across layers is essential for reconciling functional heterogeneity.

*Table 4.* Comparison between Fixed-Depth aggregation (Flat-Model Assumption) and Layer-wise Adaptive aggregation.

| Aggregation Depth | Constraint | Avg. Acc. |
|---|---|---|
| Fixed ($k = 1$) | Global Only | 87.20 |
| Fixed ($k = 4$) | Coarse Clusters | 91.45 |
| Fixed ($k = 8$) | Fine-grained | 90.74 |
| **Layer-wise Adaptive** | **Dynamic ($c_l^*$)** | **92.36** |

*(3) Importance of Global Structural Consistency.* Finally, we examine the necessity of constructing a global dependency tree (Sec. 5.1) versus performing *Independent Layer-wise Clustering*. In the latter, clients are clustered at each layer based solely on that layer's local distance matrix $D^{(l)}$, without enforcing the topological constraints of the global tree. Table 5 shows that independent clustering degrades performance. We attribute this to *topological incoherence*, where client groupings fluctuate chaotically between adjacent layers, disrupting the semantic continuity of the for-

ward pass. By anchoring decisions to a global skeleton, FedTreeLoRA ensures smooth transitions in expert specialization, which is critical for stable fine-tuning.

*Table 5.* Impact of the Global Tree Skeleton on performance.

| Structural Prior | Consistency | Avg. Acc. |
|---|---|---|
| Independent Clustering | Low | 89.47 |
| **Global Tree Skeleton** | **High** | **92.36** |

**Sensitivity Analysis.** We further conduct comprehensive sensitivity analyses to evaluate the robustness of FedTreeLoRA under varying system configurations, including the number of local epochs, LoRA rank, client population size, and degrees of data heterogeneity. We also analyze the impact of our framework-specific hyperparameters: the heterogeneity threshold $\tau$ and search range $K$. The detailed results and discussions are provided in Appendix C.

## 6.2. Natural Language Generation

To validate the generalizability of FedTreeLoRA beyond classification tasks, we evaluate its performance on Natural Language Generation (NLG) benchmarks.

**NLG Setup.** We employ LLaMA-2-7B (Touvron et al., 2023a), quantized to 8-bit precision, as the base model. To construct a realistic heterogeneous federated setting, we utilize four diverse datasets from the FLAN collection (Chung et al., 2024): *Text Editing*, *Struct to Text*, *Sentiment Analysis*, and *Commonsense Reasoning*. Unlike NLU tasks where heterogeneity is often modeled via label skew, NLG tasks exhibit inherent functional diversity. Therefore, we adopt the task-heterogeneous setup proposed in (Wang et al., 2025): a total of $N = 8$ clients are partitioned such that each dataset is assigned to two specific clients. Each client possesses 600 training samples and 200 test samples. Training is conducted using the AdamW optimizer with a batch size of 8 for $E = 2$ local epochs over $T = 10$ communication rounds. We apply LoRA adapters with rank $r = 8$ to the query and value projections. Learning rates are tuned via grid search over $\eta \in \{1E-4, 3E-4, 1E-3, 3E-3, 1E-2\}$. Following standard evaluation protocols (Yang et al., 2024), we report ROUGE-1 scores as the primary metric.

**Performance Comparison.** Table 2 summarizes the results on the NLG benchmarks. Consistent with our NLU findings, FedTreeLoRA achieves the best performance across all four generation tasks, with a parameter budget comparable to FedIT. It outperforms the strongest baselines by a clear margin despite using fewer trainable parameters, with particularly strong gains on tasks requiring structured planning and semantic reasoning (e.g., *Text Editing*). This confirms that our hierarchical aggregation strategy effectively preserves the delicate generative capabilities required for diverse tasks while mitigating the interference typically caused by aggre-

gating conflicting domains. Additional results on a model from an alternative LLM family are reported in Appendix F.

# 7. Conclusion

In this paper, we challenge the conventional 'Flat-Model Assumption' in federated fine-tuning and propose FedTreeLoRA, a framework that reconciles statistical and functional heterogeneity via tree-structured aggregation. By aligning the aggregation granularity with the layer-wise hierarchy of LLMs, sharing shallow linguistic features while personalizing deep semantic reasoning, FedTreeLoRA consistently outperforms state-of-the-art methods across diverse NLU and NLG benchmarks. We achieve these gains with negligible parameter overhead, empirically validating that topological precision is more effective than indiscriminate capacity expansion. Our work offers a new perspective on efficiently scaling personalized federated fine-tuning.

# Acknowledgments

The work of Jieming Bian, Lei Wang and Jie Xu is partially supported by NSF under grants 2433886, 2505381 and 2515982. The work of Letian Zhang is partially supported by NSF under grant 2348279 and also supported by MTSU Stark Land project.

# Impact Statement

This paper presents work whose goal is to advance the field of Machine Learning. There are many potential societal consequences of our work, none which we feel must be specifically highlighted here.

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

# Contents of Appendix

This appendix contains the following sections:

# A. Convergence Analysis: Assumptions and Proof

In this appendix we formalize the assumptions used in Section 5.4 and provide a complete proof of Theorem 5.1. We recall that the effective weights of client $k$ at communication round $t$ are denoted by

$$\mathbf{W}_k^{(t)} = \{W_{l,k}^{(t)}\}_{l=1}^{L},$$

and for each layer $l$ the forward pass (cf. Eq. (8)) is

$$h_l(x) = W_{0,l}x + \lambda_{l,k}^{(t)} \bar{B}_{l,k}^{\text{clus},(t)} \bar{A}_{l,k}^{\text{clus},(t)} x + \left(1 - \lambda_{l,k}^{(t)}\right) \bar{B}_{l,k}^{\text{ext},(t)} \bar{A}_{l,k}^{\text{ext},(t)} x. \tag{9}$$

During local training in round $t$, each client $k$ updates only its Cluster Expert $\left(\bar{A}_{l,k}^{\text{clus},(t)}, \bar{B}_{l,k}^{\text{clus},(t)}\right)$ and mixing coefficient $\lambda_{l,k}^{(t)}$, while the External Expert $\left(\bar{A}_{l,k}^{\text{ext},(t)}, \bar{B}_{l,k}^{\text{ext},(t)}\right)$ is frozen; the experts themselves are obtained by averaging client-level LoRA factors according to the tree-structured aggregation in Eq. (7).

## A.1. Assumptions

We adopt standard smoothness and stochastic gradient assumptions from federated optimization, together with a LoRA-specific boundedness and alignment condition that is in line with the formulation of Guo et al. (2025).

**Assumption A.1** ($\sigma$-smoothness). Each local objective $\mathcal{L}_k$ is $\sigma$-smooth with respect to the adapted weights. That is, for any two collections of weights $\mathbf{W}, \mathbf{W}'$,

$$\mathcal{L}_k(\mathbf{W}') \leq \mathcal{L}_k(\mathbf{W}) + \left\langle \mathbf{W}' - \mathbf{W}, \nabla \mathcal{L}_k(\mathbf{W}) \right\rangle_F + \frac{\sigma}{2} \|\mathbf{W}' - \mathbf{W}\|_F^2.$$

**Assumption A.2** (Unbiased stochastic gradients with bounded variance). Let $\xi_{k,t,e}$ be sampled uniformly from client $k$'s local dataset at local step $e$ in communication round $t$, and let $\mathbf{W}_k^{(t,e)}$ denote the local model at that step (with $\mathbf{W}_k^{(t,0)} = \mathbf{W}_k^{(t)}$). Then

$$\mathbb{E}_{\xi_{k,t,e}}\left[\nabla \mathcal{L}_k(\mathbf{W}_k^{(t,e)}; \xi_{k,t,e})\right] = \nabla \mathcal{L}_k(\mathbf{W}_k^{(t,e)}), \qquad \mathbb{E}_{\xi_{k,t,e}}\left[\left\|\nabla \mathcal{L}_k(\mathbf{W}_k^{(t,e)}; \xi_{k,t,e})\right\|_F^2\right] \leq G^2,$$

for some constant $G > 0$.

**Assumption A.3** (LoRA parameter bounds and alignment). There exist constants $M_A, M_B > 0$ and $\mu_A, \mu_B > 0$ such that, for all clients $k$, layers $l$, and rounds $t$, the underlying client-level LoRA factors $A_{l,k}^{(t)}$ and $B_{l,k}^{(t)}$ satisfy

$$\left\|A_{l,k}^{(t)}\right\|_F \leq M_A, \qquad \left\|B_{l,k}^{(t)}\right\|_F \leq M_B.$$

Let $G_{l,k}^{(t)} := \nabla_{W_{l,k}} \mathcal{L}_k(\mathbf{W}_k^{(t)})$ denote the layer-wise gradient at the adapted weights $\mathbf{W}_k^{(t)}$. We assume the following alignment inequalities hold:

$$\sum_{l=1}^{L} \left\langle \left(A_{l,k}^{(t)}\right)^{\top} A_{l,k}^{(t)}, \left(G_{l,k}^{(t)}\right)^{\top} G_{l,k}^{(t)} \right\rangle_F \geq \mu_A \left\|\nabla \mathcal{L}_k(\mathbf{W}_k^{(t)})\right\|_F^2, \tag{10}$$

$$\sum_{l=1}^{L} \left\langle B_{l,k}^{(t)} \left(B_{l,k}^{(t)}\right)^{\top}, G_{l,k}^{(t)} \left(G_{l,k}^{(t)}\right)^{\top} \right\rangle_F \geq \mu_B \left\|\nabla \mathcal{L}_k(\mathbf{W}_k^{(t)})\right\|_F^2. \tag{11}$$

In FedTreeLoRA, both the Cluster Expert and the External Expert are constructed as (weighted) averages of the client-level factors $\{A_{l,j}^{(t)}, B_{l,j}^{(t)}\}_j$ over appropriate subsets defined by the tree structure (cf. Eq. (7)). Therefore, they inherit the same Frobenius-norm bounds and alignment properties, possibly with smaller constants, which we absorb into $M_A, M_B, \mu_A, \mu_B$ for notational simplicity. Furthermore, the mixing coefficients satisfy $\lambda_{l,k}^{(t)} \in [0, 1]$ for all $(l, k, t)$.

Assumptions A.1 and A.2 are standard in non-convex federated optimization, while Assumption A.3 follows the low-rank alignment formulation of Guo et al. (2025), adapted to the expert-based aggregation of FedTreeLoRA.

## A.2. Proof of Theorem 5.1

We now prove the convergence result stated in Theorem 5.1. The proof follows the standard "one-round descent + telescoping" structure, but tailored to the low-rank updates and expert mixing in FedTreeLoRA.

**Notation within one round.** Fix a communication round $t$. We index local steps by $e \in \{0, 1, \ldots, E\}$ and write

$$\mathbf{W}_k^{(t,0)} := \mathbf{W}_k^{(t)}, \qquad \mathbf{W}_k^{(t,E)} := \mathbf{U}_k^{(t)},$$

for the model at the beginning and after the $E$ local updates, respectively. For client $k$ and layer $l$, we denote by

$$A_{l,k}^{(t,e)}, \quad B_{l,k}^{(t,e)}, \quad \lambda_{l,k}^{(t,e)}$$

the trainable low-rank factors and mixing coefficient at step $e$. These correspond to the Cluster Expert and its mixing weight in Eq. (9); the External Expert remains fixed throughout the round and is thus not indexed by $e$.

Let

$$G_{l,k}^{(t,e)} := \nabla_{W_{l,k}} \mathcal{L}_k(\mathbf{W}_k^{(t,e)}; \xi_{k,t,e}),$$

and denote the full gradient at the beginning of the round by

$$\nabla \mathcal{L}_k(\mathbf{W}_k^{(t)}) := \{G_{l,k}^{(t,0)}\}_{l=1}^{L}.$$

**Stage 1: descent during local updates.** Using Eq. (9) and the chain rule, the stochastic gradients with respect to the low-rank factors and mixing coefficient at step $e$ can be written as

$$\nabla_{B_{l,k}} \mathcal{L}_k(\mathbf{W}_k^{(t,e)}; \xi_{k,t,e}) = \lambda_{l,k}^{(t,e)} G_{l,k}^{(t,e)} (A_{l,k}^{(t,e)})^{\top}, \tag{12}$$

$$\nabla_{A_{l,k}} \mathcal{L}_k(\mathbf{W}_k^{(t,e)}; \xi_{k,t,e}) = \lambda_{l,k}^{(t,e)} (B_{l,k}^{(t,e)})^{\top} G_{l,k}^{(t,e)}, \tag{13}$$

$$\nabla_{\lambda_{l,k}} \mathcal{L}_k(\mathbf{W}_k^{(t,e)}; \xi_{k,t,e}) = \langle B_{l,k}^{(t,e)} A_{l,k}^{(t,e)} - \bar{B}_{l,k}^{\text{ext},(t)} \bar{A}_{l,k}^{\text{ext},(t)}, G_{l,k}^{(t,e)} \rangle_F. \tag{14}$$

The SGD updates with stepsize $\eta > 0$ are

$$B_{l,k}^{(t,e+1)} = B_{l,k}^{(t,e)} - \eta \nabla_{B_{l,k}} \mathcal{L}_k(\mathbf{W}_k^{(t,e)}; \xi_{k,t,e}), \tag{15}$$

$$A_{l,k}^{(t,e+1)} = A_{l,k}^{(t,e)} - \eta \nabla_{A_{l,k}} \mathcal{L}_k(\mathbf{W}_k^{(t,e)}; \xi_{k,t,e}), \tag{16}$$

$$\lambda_{l,k}^{(t,e+1)} = \lambda_{l,k}^{(t,e)} - \eta \nabla_{\lambda_{l,k}} \mathcal{L}_k(\mathbf{W}_k^{(t,e)}; \xi_{k,t,e}). \tag{17}$$

Summing (15) over $e$ and applying the triangle inequality, together with Assumptions A.2 and A.3 (boundedness of $A_{l,k}^{(t,e)}$ and $B_{l,k}^{(t,e)}$), we obtain the incremental bounds

$$\left\| B_{l,k}^{(t,E)} - B_{l,k}^{(t,0)} \right\|_F \leq \eta \sum_{e=0}^{E-1} \left\| G_{l,k}^{(t,e)} (A_{l,k}^{(t,e)})^{\top} \right\|_F \leq \eta E M_A G, \tag{18}$$

$$\left\| A_{l,k}^{(t,E)} - A_{l,k}^{(t,0)} \right\|_F \leq \eta \sum_{e=0}^{E-1} \left\| (B_{l,k}^{(t,e)})^{\top} G_{l,k}^{(t,e)} \right\|_F \leq \eta E M_B G. \tag{19}$$

Similarly, using (14) and the product bound $\left\| B_{l,k}^{(t,e)} A_{l,k}^{(t,e)} - \bar{B}_{l,k}^{\text{ext},(t)} \bar{A}_{l,k}^{\text{ext},(t)} \right\|_F \leq 2 M_A M_B$ (from Assumption A.3 and the construction of the experts), we obtain

$$\left| \lambda_{l,k}^{(t,E)} - \lambda_{l,k}^{(t,0)} \right| \leq \eta E \cdot 2 M_A M_B G. \tag{20}$$

We now relate the change in the model parameters to the gradient of $\mathcal{L}_k$. By $\sigma$-smoothness (Assumption A.1), we have

$$\mathcal{L}_k(\mathbf{U}_k^{(t)}) \leq \mathcal{L}_k(\mathbf{W}_k^{(t)}) + \langle \mathbf{U}_k^{(t)} - \mathbf{W}_k^{(t)}, \nabla \mathcal{L}_k(\mathbf{W}_k^{(t)}) \rangle_F + \frac{\sigma}{2} \left\| \mathbf{U}_k^{(t)} - \mathbf{W}_k^{(t)} \right\|_F^2. \tag{21}$$

The perturbation $\mathbf{U}_k^{(t)} - \mathbf{W}_k^{(t)}$ is induced by the cumulative changes in $\{A_{l,k}^{(t,e)}, B_{l,k}^{(t,e)}, \lambda_{l,k}^{(t,e)}\}_{l,e}$. Using (18)–(20), the submultiplicativity of the Frobenius norm, and the fact that $|\lambda_{l,k}^{(t,e)}| \leq 1$, a standard calculation shows that

$$\mathbb{E}\left[\|\mathbf{U}_k^{(t)} - \mathbf{W}_k^{(t)}\|_F^2\right] \leq 24 L \eta^2 E^2 G^2 \left(M_A^4 + M_B^4 + M_A^4 M_B^4\right). \tag{22}$$

To bound the inner-product term in (21), we compare the actual trajectory with a "reference" gradient step in the low-rank subspace at the beginning of the round. Define

$$Z_{l,k}^{A,(t)} := G_{l,k}^{(t,0)}\left(A_{l,k}^{(t,0)}\right)^\top, \qquad Z_{l,k}^{B,(t)} := \left(B_{l,k}^{(t,0)}\right)^\top G_{l,k}^{(t,0)}.$$

Using the Frobenius trace identity $\|XY^\top\|_F^2 = \langle Y^\top Y, X^\top X\rangle_F$ together with the alignment conditions (10)–(11), we obtain

$$\sum_{l=1}^{L} \|Z_{l,k}^{A,(t)}\|_F^2 = \sum_{l=1}^{L} \langle\left(A_{l,k}^{(t,0)}\right)^\top A_{l,k}^{(t,0)}, \left(G_{l,k}^{(t,0)}\right)^\top G_{l,k}^{(t,0)}\rangle_F \geq \mu_A \|\nabla\mathcal{L}_k(\mathbf{W}_k^{(t)})\|_F^2, \tag{23}$$

$$\sum_{l=1}^{L} \|Z_{l,k}^{B,(t)}\|_F^2 = \sum_{l=1}^{L} \langle B_{l,k}^{(t,0)}\left(B_{l,k}^{(t,0)}\right)^\top, G_{l,k}^{(t,0)}\left(G_{l,k}^{(t,0)}\right)^\top\rangle_F \geq \mu_B \|\nabla\mathcal{L}_k(\mathbf{W}_k^{(t)})\|_F^2. \tag{24}$$

Following the same variance-decomposition argument as in Guo et al. (2025), one can show that the cumulative effect of the SGD updates (15)–(17) yields

$$\mathbb{E}\left[\langle\mathbf{U}_k^{(t)} - \mathbf{W}_k^{(t)}, \nabla\mathcal{L}_k(\mathbf{W}_k^{(t)})\rangle_F\right] \leq -\eta(\mu_A + \mu_B) E\mathbb{E}\left[\|\nabla\mathcal{L}_k(\mathbf{W}_k^{(t)})\|_F^2\right] + \eta^2 E^2 L M_A^2 M_B^2 G^2. \tag{25}$$

The negative term comes from the aligned reference step quantified by (23)–(24), while the $O(\eta^2)$ term collects the higher-order deviations caused by (i) evaluating gradients at intermediate points $\mathbf{W}_k^{(t,e)}$ and (ii) the updates of the mixing coefficients $\lambda_{l,k}^{(t,e)}$.

Substituting (22) and (25) into (21), and grouping the $O(\eta^2)$ terms, we obtain the Stage-1 descent inequality

$$\begin{aligned}
\mathbb{E}[\mathcal{L}_k(U_k^{(t)})] \leq \mathbb{E}[\mathcal{L}_k(W_k^{(t)})] &- \eta(\mu_A + \mu_B) E\mathbb{E}[\|\nabla\mathcal{L}_k(W_k^{(t)})\|_F^2] \\
&+ \eta^2 E^2 L M_A^2 M_B^2 G^2 + 12\sigma L \eta^2 E^2 G^2 (M_A^4 + M_B^4 + M_A^4 M_B^4).
\end{aligned} \tag{26}$$

**Stage 2: tree-structured aggregation.** We now make precise the effect of the tree-structured aggregation: (i) the mismatch between a client's locally updated Cluster Expert and the aggregated Cluster Expert shared by all peers in the same node; and (ii) the change in the External Expert induced by the peers outside the node.

Fix a round $t$, a client $k$, and a layer $l$. After the $E$ local updates in Stage 1, the layer used by client $k$ has the form

$$U_{l,k}^{(t)} = W_{0,l} + \lambda_{l,k}^{(t,E)} B_{l,k}^{(t,E)} A_{l,k}^{(t,E)} + \left(1 - \lambda_{l,k}^{(t,E)}\right) \bar{B}_{l,k}^{\text{ext},(t)} \bar{A}_{l,k}^{\text{ext},(t)}, \tag{27}$$

where $A_{l,k}^{(t,E)}, B_{l,k}^{(t,E)}, \lambda_{l,k}^{(t,E)}$ are the locally updated Cluster Expert and mixing coefficient, and $\bar{A}_{l,k}^{\text{ext},(t)}, \bar{B}_{l,k}^{\text{ext},(t)}$ is the (frozen) External Expert from the previous round.

Let $\mathcal{S}_k^{(l)}$ denote the set of clients that share the same tree node as $k$ at layer $l$, and $\mathcal{R}_k^{(l)}$ its complement (cf. Section 5.3). Tree-structured aggregation at the end of round $t$ updates the experts as follows (cf. Eq. (7)):

$$\begin{aligned}
\bar{A}_{l,k}^{\text{clus},(t+1)} &= \frac{1}{|\mathcal{S}_k^{(l)}|} \sum_{j \in \mathcal{S}_k^{(l)}} A_{l,j}^{(t,E)}, \qquad \bar{B}_{l,k}^{\text{clus},(t+1)} = \frac{1}{|\mathcal{S}_k^{(l)}|} \sum_{j \in \mathcal{S}_k^{(l)}} B_{l,j}^{(t,E)}, \\
\bar{A}_{l,k}^{\text{ext},(t+1)} &= \frac{1}{|\mathcal{R}_k^{(l)}|} \sum_{j \in \mathcal{R}_k^{(l)}} A_{l,j}^{(t,E)}, \qquad \bar{B}_{l,k}^{\text{ext},(t+1)} = \frac{1}{|\mathcal{R}_k^{(l)}|} \sum_{j \in \mathcal{R}_k^{(l)}} B_{l,j}^{(t,E)}.
\end{aligned} \tag{28}$$

The post-aggregation layer used by client $k$ at layer $l$ in the next round is

$$W_{l,k}^{(t+1)} = W_{0,l} + \lambda_{l,k}^{(t,E)} \bar{B}_{l,k}^{\text{clus},(t+1)} \bar{A}_{l,k}^{\text{clus},(t+1)} + \left(1 - \lambda_{l,k}^{(t,E)}\right) \bar{B}_{l,k}^{\text{ext},(t+1)} \bar{A}_{l,k}^{\text{ext},(t+1)}. \tag{29}$$

Subtracting (27) from (29), we obtain the decomposition

$$W_{l,k}^{(t+1)} - U_{l,k}^{(t)} = \underbrace{\lambda_{l,k}^{(t,E)} \left(\bar{B}_{l,k}^{\text{clus},(t+1)} \bar{A}_{l,k}^{\text{clus},(t+1)} - B_{l,k}^{(t,E)} A_{l,k}^{(t,E)}\right)}_{\text{(i) Cluster Expert aggregation difference}}$$

$$+ \underbrace{\left(1 - \lambda_{l,k}^{(t,E)}\right) \left(\bar{B}_{l,k}^{\text{ext},(t+1)} \bar{A}_{l,k}^{\text{ext},(t+1)} - \bar{B}_{l,k}^{\text{ext},(t)} \bar{A}_{l,k}^{\text{ext},(t)}\right)}_{\text{(ii) External Expert update difference}}. \tag{30}$$

We now bound the two terms separately in Frobenius norm.

**(i) Cluster Expert aggregation difference.** Fix $l$ and $\mathcal{S}_k^{(l)}$. By construction of FedTreeLoRA, all clients $j \in \mathcal{S}_k^{(l)}$ share the *same* Cluster Expert at the *beginning* of round $t$:

$$A_{l,j}^{(t,0)} = A_{l,k}^{(t,0)}, \qquad B_{l,j}^{(t,0)} = B_{l,k}^{(t,0)} \quad \forall j \in \mathcal{S}_k^{(l)}.$$

After $E$ local updates, we can write

$$A_{l,j}^{(t,E)} = A_{l,k}^{(t,0)} + \Delta A_{l,j}^{(t)}, \qquad B_{l,j}^{(t,E)} = B_{l,k}^{(t,0)} + \Delta B_{l,j}^{(t)},$$

with $\|\Delta A_{l,j}^{(t)}\|_F \le \eta E M_B G$ and $\|\Delta B_{l,j}^{(t)}\|_F \le \eta E M_A G$ for all $j$ by (19)–(18). In particular, $A_{l,k}^{(t,E)} = A_{l,k}^{(t,0)} + \Delta A_{l,k}^{(t)}$, $B_{l,k}^{(t,E)} = B_{l,k}^{(t,0)} + \Delta B_{l,k}^{(t)}$. From (28), the aggregated Cluster Expert can be expressed as

$$\bar{A}_{l,k}^{\text{clus},(t+1)} = A_{l,k}^{(t,0)} + \frac{1}{|\mathcal{S}_k^{(l)}|} \sum_{j \in \mathcal{S}_k^{(l)}} \Delta A_{l,j}^{(t)}, \qquad \bar{B}_{l,k}^{\text{clus},(t+1)} = B_{l,k}^{(t,0)} + \frac{1}{|\mathcal{S}_k^{(l)}|} \sum_{j \in \mathcal{S}_k^{(l)}} \Delta B_{l,j}^{(t)}.$$

Therefore,

$$\bar{A}_{l,k}^{\text{clus},(t+1)} - A_{l,k}^{(t,E)} = \frac{1}{|\mathcal{S}_k^{(l)}|} \sum_{j \in \mathcal{S}_k^{(l)}} \Delta A_{l,j}^{(t)} - \Delta A_{l,k}^{(t)}, \tag{31}$$

$$\bar{B}_{l,k}^{\text{clus},(t+1)} - B_{l,k}^{(t,E)} = \frac{1}{|\mathcal{S}_k^{(l)}|} \sum_{j \in \mathcal{S}_k^{(l)}} \Delta B_{l,j}^{(t)} - \Delta B_{l,k}^{(t)}. \tag{32}$$

Using the triangle inequality and the bounds on $\Delta A_{l,j}^{(t)}, \Delta B_{l,j}^{(t)}$, we obtain

$$\left\|\bar{A}_{l,k}^{\text{clus},(t+1)} - A_{l,k}^{(t,E)}\right\|_F \le \frac{1}{|\mathcal{S}_k^{(l)}|} \sum_{j \in \mathcal{S}_k^{(l)}} \left\|\Delta A_{l,j}^{(t)}\right\|_F + \left\|\Delta A_{l,k}^{(t)}\right\|_F \le 2\eta E M_B G, \tag{33}$$

$$\left\|\bar{B}_{l,k}^{\text{clus},(t+1)} - B_{l,k}^{(t,E)}\right\|_F \le 2\eta E M_A G. \tag{34}$$

Moreover, since each $A_{l,j}^{(t,E)}, B_{l,j}^{(t,E)}$ remains inside the Frobenius ball of radius $M_A, M_B$ (Assumption A.3), their averages also satisfy

$$\left\|\bar{A}_{l,k}^{\text{clus},(t+1)}\right\|_F \le M_A, \qquad \left\|\bar{B}_{l,k}^{\text{clus},(t+1)}\right\|_F \le M_B.$$

We now bound the difference between the two Cluster-Expert products. Using $\bar{B}\bar{A} - BA = \bar{B}(\bar{A} - A) + (\bar{B} - B)A$ and the submultiplicativity of the Frobenius norm, we have

$$\left\|\bar{B}_{l,k}^{\text{clus},(t+1)} \bar{A}_{l,k}^{\text{clus},(t+1)} - B_{l,k}^{(t,E)} A_{l,k}^{(t,E)}\right\|_F \le \left\|\bar{B}_{l,k}^{\text{clus},(t+1)}\right\|_F \left\|\bar{A}_{l,k}^{\text{clus},(t+1)} - A_{l,k}^{(t,E)}\right\|_F + \left\|\bar{B}_{l,k}^{\text{clus},(t+1)} - B_{l,k}^{(t,E)}\right\|_F \left\|A_{l,k}^{(t,E)}\right\|_F$$

$$\le M_B \cdot 2\eta E M_B G + M_A \cdot 2\eta E M_A G$$

$$= 2\eta E G \left(M_A^2 + M_B^2\right). \tag{35}$$

Multiplying by $|\lambda_{l,k}^{(t,E)}| \leq 1$ and using $(a+b)^2 \leq 2(a^2+b^2)$, we deduce that the squared norm of the Cluster Expert aggregation difference in (30) is bounded by

$$\left\| \lambda_{l,k}^{(t,E)} \left( \bar{B}_{l,k}^{\text{clus},(t+1)} \bar{A}_{l,k}^{\text{clus},(t+1)} - B_{l,k}^{(t,E)} A_{l,k}^{(t,E)} \right) \right\|_F^2 \leq 8\eta^2 E^2 G^2 \left( M_A^4 + M_B^4 \right), \tag{36}$$

up to an absolute numerical constant.

**(ii) External Expert update difference.** We now turn to the second term in (30). At the *beginning* of round $t$, all clients in $\mathcal{R}_k^{(l)}$ share the same External Expert $\bar{A}_{l,k}^{\text{ext},(t)}, \bar{B}_{l,k}^{\text{ext},(t)}$, which is a fixed average of Cluster Experts from the previous round. During local training in round $t$, this External Expert is kept frozen, and only the Cluster Experts $\{A_{l,j}^{(t,e)}, B_{l,j}^{(t,e)}\}$ are updated. At the end of the round, the new External Expert is defined in (28) as an average of the *updated* Cluster Experts of clients in $\mathcal{R}_k^{(l)}$:

$$\bar{A}_{l,k}^{\text{ext},(t+1)} = \frac{1}{|\mathcal{R}_k^{(l)}|} \sum_{j \in \mathcal{R}_k^{(l)}} A_{l,j}^{(t,E)}, \qquad \bar{B}_{l,k}^{\text{ext},(t+1)} = \frac{1}{|\mathcal{R}_k^{(l)}|} \sum_{j \in \mathcal{R}_k^{(l)}} B_{l,j}^{(t,E)}.$$

Writing again $A_{l,j}^{(t,E)} = A_{l,j}^{(t,0)} + \Delta A_{l,j}^{(t)}, B_{l,j}^{(t,E)} = B_{l,j}^{(t,0)} + \Delta B_{l,j}^{(t)}$ and using the fact that $\bar{A}_{l,k}^{\text{ext},(t)}$ and $\bar{B}_{l,k}^{\text{ext},(t)}$ are averages of $\{A_{l,j}^{(t,0)}, B_{l,j}^{(t,0)}\}_{j \in \mathcal{R}_k^{(l)}}$, we can write

$$\bar{A}_{l,k}^{\text{ext},(t+1)} - \bar{A}_{l,k}^{\text{ext},(t)} = \frac{1}{|\mathcal{R}_k^{(l)}|} \sum_{j \in \mathcal{R}_k^{(l)}} \Delta A_{l,j}^{(t)}, \qquad \bar{B}_{l,k}^{\text{ext},(t+1)} - \bar{B}_{l,k}^{\text{ext},(t)} = \frac{1}{|\mathcal{R}_k^{(l)}|} \sum_{j \in \mathcal{R}_k^{(l)}} \Delta B_{l,j}^{(t)}.$$

By (19)–(18),

$$\left\| \bar{A}_{l,k}^{\text{ext},(t+1)} - \bar{A}_{l,k}^{\text{ext},(t)} \right\|_F \leq \eta E M_B G, \qquad \left\| \bar{B}_{l,k}^{\text{ext},(t+1)} - \bar{B}_{l,k}^{\text{ext},(t)} \right\|_F \leq \eta E M_A G, \tag{37}$$

and both External Experts remain bounded: $\left\| \bar{A}_{l,k}^{\text{ext},(t)} \right\|_F, \left\| \bar{A}_{l,k}^{\text{ext},(t+1)} \right\|_F \leq M_A, \left\| \bar{B}_{l,k}^{\text{ext},(t)} \right\|_F, \left\| \bar{B}_{l,k}^{\text{ext},(t+1)} \right\|_F \leq M_B$.

Analogously to (35), we obtain

$$\left\| \bar{B}_{l,k}^{\text{ext},(t+1)} \bar{A}_{l,k}^{\text{ext},(t+1)} - \bar{B}_{l,k}^{\text{ext},(t)} \bar{A}_{l,k}^{\text{ext},(t)} \right\|_F \leq M_B \cdot \eta E M_B G + M_A \cdot \eta E M_A G$$
$$= \eta E G \left( M_A^2 + M_B^2 \right), \tag{38}$$

and therefore

$$\left\| \left( 1 - \lambda_{l,k}^{(t,E)} \right) \left( \bar{B}_{l,k}^{\text{ext},(t+1)} \bar{A}_{l,k}^{\text{ext},(t+1)} - \bar{B}_{l,k}^{\text{ext},(t)} \bar{A}_{l,k}^{\text{ext},(t)} \right) \right\|_F^2 \leq 2\eta^2 E^2 G^2 \left( M_A^4 + M_B^4 \right), \tag{39}$$

again up to an absolute constant.

**Combining (i) and (ii).** By (30), the per-layer difference satisfies

$$\left\| W_{l,k}^{(t+1)} - U_{l,k}^{(t)} \right\|_F^2 \leq 2 \left\| (\text{i}) \right\|_F^2 + 2 \left\| (\text{ii}) \right\|_F^2,$$

so that, combining (36) and (39),

$$\left\| W_{l,k}^{(t+1)} - U_{l,k}^{(t)} \right\|_F^2 \leq C_0 \, \eta^2 E^2 G^2 \left( M_A^4 + M_B^4 \right), \tag{40}$$

for some universal constant $C_0 > 0$. Summing over $l = 1, \ldots, L$ and taking expectation, we obtain the aggregate bound

$$\mathbb{E} \left[ \left\| \mathbf{W}_k^{(t+1)} - \mathbf{U}_k^{(t)} \right\|_F^2 \right] \leq C_0 L \eta^2 E^2 G^2 \left( M_A^4 + M_B^4 \right), \tag{41}$$

Applying Assumption A.1 at $\mathbf{U}_k^{(t)}$, setting the learning rate $\eta = O(\frac{1}{EL})$ and using Cauchy–Schwarz to bound the corresponding inner-product term (41), we obtain

$$\mathbb{E} \left[ \mathcal{L}_k(\mathbf{W}_k^{(t+1)}) \right] \leq \mathbb{E} \left[ \mathcal{L}_k(\mathbf{U}_k^{(t)}) \right] + \tilde{C} \sigma \eta^2 E^2 L G^2 \left( M_A^4 + M_B^4 \right), \tag{42}$$

for some absolute constant $\tilde{C} > 0$.

**One-round recursion and telescoping.** Combining the Stage 1 descent inequality (cf. Eq. (26)) with the Stage 2 aggregation bound (42), we obtain the following one-round recursion for each client $k$ and communication round $t$:

$$\mathbb{E}\big[\mathcal{L}_k(\mathbf{W}_k^{(t+1)})\big] \ \leq \ \mathbb{E}\big[\mathcal{L}_k(\mathbf{W}_k^{(t)})\big] - \eta(\mu_A + \mu_B)E\mathbb{E}\big[\|\nabla\mathcal{L}_k(\mathbf{W}_k^{(t)})\|_F^2\big] + \eta^2 E^2\Gamma, \tag{43}$$

where $\Gamma$ is given by

$$\Gamma := LM_A^2 M_B^2 G^2 + C\,\sigma L G^2\big(M_A^4 + M_B^4 + M_A^4 M_B^4\big), \tag{44}$$

for some absolute constant $C > 0$.

Summing (43) over $t = 0, \ldots, T-1$ and using telescoping yields

$$\sum_{t=0}^{T-1}\mathbb{E}\big[\|\nabla\mathcal{L}_k(\mathbf{W}_k^{(t)})\|_F^2\big] \ \leq \ \frac{\Delta_k}{\eta E(\mu_A + \mu_B)} + \frac{\eta E\Gamma T}{\mu_A + \mu_B}, \tag{45}$$

where $\Delta_k := \mathcal{L}_k(\mathbf{W}_k^{(0)}) - \mathcal{L}_k(\mathbf{W}_k^\star)$ is the initial function gap for client $k$. Let $\Delta \geq \max_k \Delta_k$, sum over all $k$, and divide both sides by $NT$ to obtain

$$\frac{1}{NT}\sum_{k=1}^{N}\sum_{t=0}^{T-1}\mathbb{E}\big[\|\nabla\mathcal{L}_k(\mathbf{W}_k^{(t)})\|_F^2\big] \ \leq \ \frac{\Delta}{\eta E(\mu_A + \mu_B)} + \frac{\eta E\Gamma}{\mu_A + \mu_B}. \tag{46}$$

Choosing $\eta E = \sqrt{\Delta/(\Gamma T)}$ minimizes the right-hand side of (46) and yields

$$\frac{1}{NT}\sum_{k=1}^{N}\sum_{t=1}^{T}\mathbb{E}\big[\|\nabla\mathcal{L}_k(\mathbf{W}_k^{(t)})\|_F^2\big] \ \leq \ \frac{2}{\mu_A + \mu_B}\sqrt{\frac{\Delta \cdot \Gamma}{T}},$$

which coincides with the convergence guarantee stated in Theorem 5.1.

# B. Detailed Motivational Studies

In this section, we provide the comprehensive experimental setup and extended results for the motivational studies discussed in Section 4. These studies aim to further substantiate our observations regarding vertical heterogeneity and the coupling between model depth and data similarity across a broader range of tasks.

## B.1. Experimental Setup

For all motivational studies, we employ RoBERTa-Large as the backbone model. We utilize the GLUE benchmark datasets, including MNLI, QNLI, SST2 and QQP. Following standard federated settings, data is partitioned using a Dirichlet distribution $\text{Dir}(\alpha)$. To observe the impact of layer-wise aggregation, we divide the 24 Transformer layers into *Shallow* (layers 1–12) and *Deep* (layers 13–24).

## B.2. Extended Analysis of Observation 1: Impact of Functional Heterogeneity

In the main paper, Figure 1 illustrates that aggregating shallow layers is more stable than aggregating deep layers for MNLI and QNLI with 10 clients ($\alpha = 0.5$), where each client holds 1000 training samples. We extend this analysis to SST2 and QQP. As shown in Figure 5a, we observe a consistent trend across all four datasets: deep-layer aggregation frequently leads to performance degradation (negative transfer) compared to local-only training. This confirms that the functional role of shallow layers as general feature extractors is universal across diverse NLU tasks.

## B.3. Extended Analysis of Observation 2: Coupling of Dual Heterogeneity

**Rationale for the Controlled 2-Client Setup.** To rigorously isolate the coupling effect between functional depth and statistical heterogeneity, we prioritize a controlled 2-client environment over a large-scale population with stochastic Dirichlet-based partitioning (e.g., $N = 20, \alpha = 0.5$). The primary rationale is the avoidance of *spurious similarities* inherent in random sampling. In a multi-client Dirichlet setting, even under low global $\alpha$, sub-groups of clients may accidentally share near-identical distributions. Such stochastic interference masks the precise layer-wise divergence: if two clients are

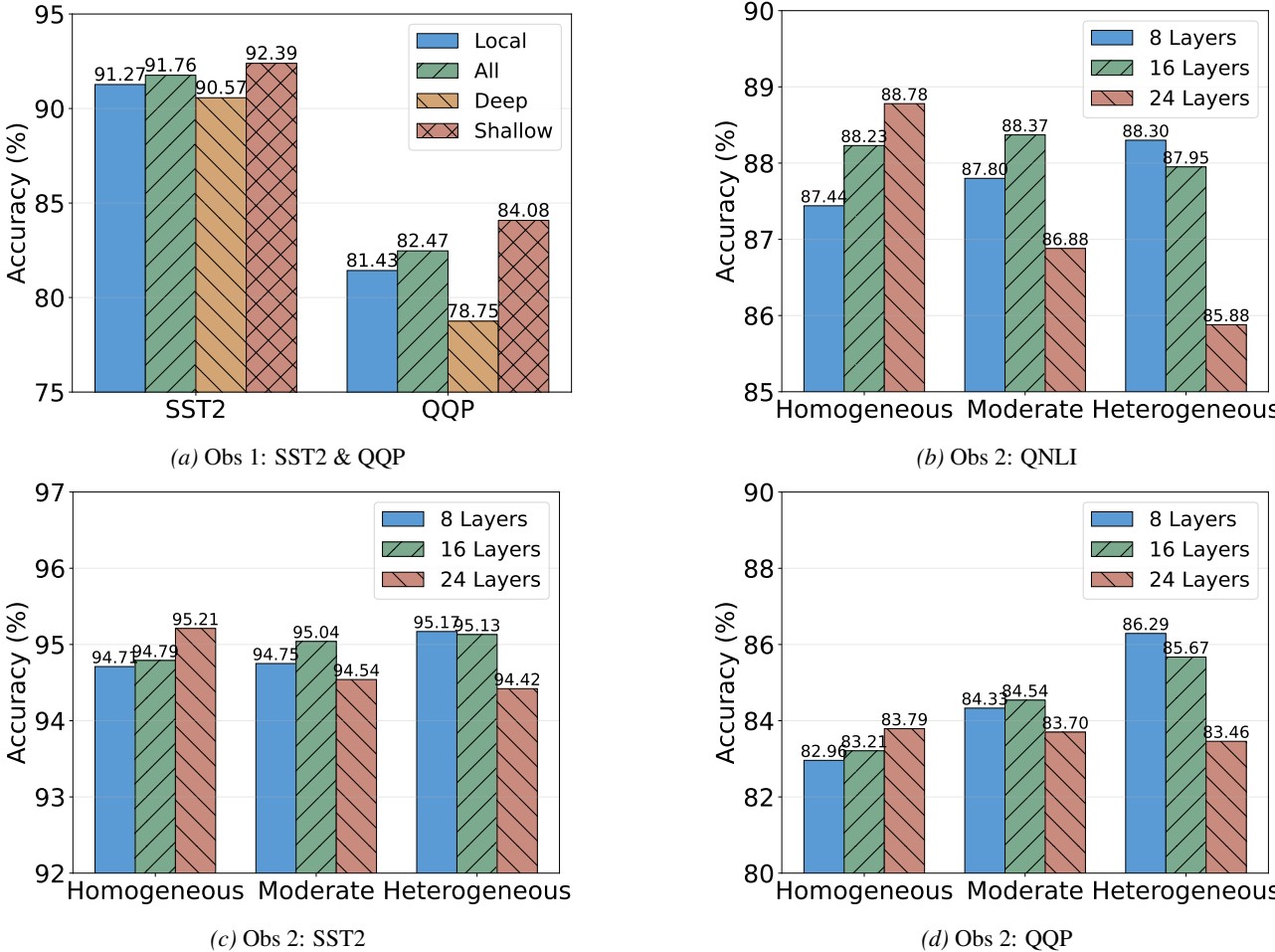

*Figure 5.* Extended Motivational Studies. (a) Substantiates *Observation 1* (Vertical Heterogeneity) on SST2 and QQP datasets. (b)–(d) Substantiate *Observation 2* (Coupling Effect) across different tasks, confirming that the optimal sharing boundary consistently shifts towards shallower layers as data heterogeneity increases.

randomly similar, separating them at shallow layers would fail to capture the true performance gap that occurs when they could have safely shared deeper parameters. By utilizing two clients with strictly defined label skew profiles, we eliminate these confounding variables and directly measure how the "safe" aggregation boundary moves in response to deterministic distribution shifts.

**Experimental Settings and Label Skew Profiles.** We conduct evaluations on RoBERTa-Large (24 layers) across three settings. For binary classification tasks (QNLI, QQP, SST2), the label distributions for Client 1 and Client 2 are defined as: *Homogeneous*: $[0.50, 0.50]$ vs. $[0.50, 0.50]$; *Moderate*: $[0.60, 0.40]$ vs. $[0.40, 0.60]$; and *Heterogeneous*: $[0.70, 0.30]$ vs. $[0.30, 0.70]$. For ternary classification task (MNLI) with distributions $[p_1, p_2, p_3]$, the settings are: *Homogeneous* with both clients sharing $[0.40, 0.40, 0.20]$; *Moderate* with Client 1: $[0.50, 0.30, 0.20]$ and Client 2: $[0.30, 0.50, 0.10]$; and *Heterogeneous* with Client 1: $[0.60, 0.20, 0.20]$ and Client 2: $[0.20, 0.60, 0.20]$.

Consistent with the findings in Figure 2, these results in Figure 5 demonstrate a systematic phase transition: as label skew intensifies, the optimal aggregation depth shifts from 24 layers toward the first 8 layers. This dataset-agnostic behavior underscores that parameter sharing must be dynamically aligned with the measured similarity between clients, a core capability of the **FedTreeLoRA** framework.

## C. Sensitivity and Robustness Analysis

In this section, we provide a comprehensive evaluation of FedTreeLoRA's robustness against variations in framework-specific hyperparameters, training configurations, and data distribution settings. Unless otherwise stated, the experimental setup follows the default configuration used in the main NLU experiments ($N = 20, \alpha = 0.5, r = 4, E = 2$).

### C.1. Impact of Framework Hyperparameters ($\tau$ and $K$)

Our framework introduces two hyperparameters that jointly govern the evolution of the aggregation tree: the heterogeneity threshold $\tau$, which regulates the resistance to splitting when deciding whether to transition from a globally shared cluster to finer partitions, and the search range $K$, which constrains how rapidly the model is allowed to refine aggregation granularity across Transformer depth.

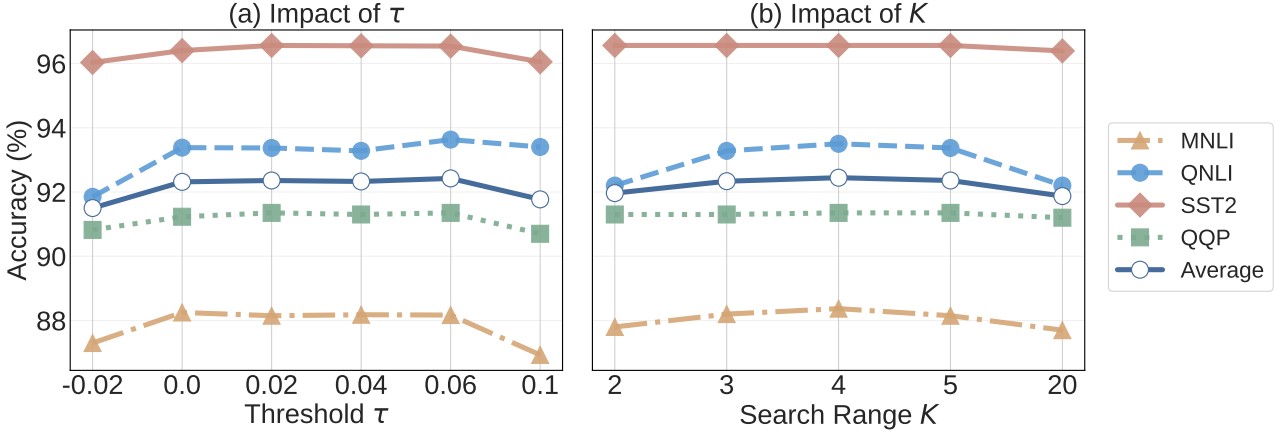

*Figure 6.* Sensitivity analysis of hyperparameters $\tau$ and $K$. FedTreeLoRA demonstrates a clear robustness plateau within $\tau \in [0, 0.06]$ and $K \in \{3, 4, 5\}$, while still outperforming all baselines even in suboptimal settings.

**Impact of Threshold $\tau$.** Recall that $\tau$ acts as the baseline score for the single-cluster case ($c = 1$), and thus determines how much statistical evidence must accumulate before a layer is allowed to branch. Empirically, the layer-wise Silhouette statistics in FedTreeLoRA naturally lie in the order of $10^{-2}$ across tasks, implying that the "effective" decision scale of $\tau$ is inherently small. Motivated by this observation, we sweep $\tau$ over a range that is already wide relative to this intrinsic scale ($\tau \in [-0.02, 0.10]$), as shown in Figure 6(a).

Within the range $\tau \in [0, 0.06]$, FedTreeLoRA exhibits a clear robustness plateau: performance remains highly stable across MNLI, QNLI, QQP, and SST-2, indicating that $\tau$ does not require fine-grained tuning. Outside this interval, the system behaves consistently with our design intuition. When $\tau < 0$, splitting is triggered too aggressively, leading to premature fragmentation and mild performance drops. When $\tau$ becomes overly conservative (e.g., $\tau = 0.10$), branching is postponed and the tree drifts toward a near-global solution, again slightly hurting accuracy. Overall, these results confirm that $\tau$ is a meaningful yet non-fragile control signal: as long as it is chosen within the natural Silhouette scale, FedTreeLoRA reliably induces appropriate hierarchical specialization.

**Impact of Search Range $K$.** Figure 6(b) analyzes the effect of the monotonicity-constrained search window $K$, which controls how smoothly the layer-wise specialization structure is allowed to evolve with depth. When $K$ is too small ($K = 2$), the framework has limited freedom to adjust aggregation structure across layers, resulting in slightly weaker performance, particularly on MNLI and QNLI. Expanding the window to a moderate range ($K \in \{3, 4, 5\}$) consistently yields the best accuracy and forms a stable robustness plateau, indicating that FedTreeLoRA benefits from controlled structural flexibility without requiring delicate tuning. Interestingly, for QQP and SST-2 the curves remain almost flat once $K \geq 2$. This behavior is fully consistent with the layer-wise structural patterns observed in Figure 4: for these datasets, the hierarchy stabilizes extremely early, typically transitioning only once from a global cluster (1) to two subgroups (2) before remaining stable. As a result, even a small search window already suffices to capture their limited specialization behavior, making them inherently less sensitive to $K$.

When $K$ becomes excessively large ($K = 20$), we observe mild degradation mainly on NLI tasks, supporting our hypothesis that overly aggressive structural jumps introduce topological incoherence and noisier specialization. Nevertheless, even under the weakest configurations ($K = 2$ or $K = 20$), FedTreeLoRA still outperforms all baseline methods in Table 1, while the moderate range $K \in \{3, 4, 5\}$ delivers the strongest and most stable improvements.

## C.2. Robustness to Training Configurations

We investigate the adaptability of FedTreeLoRA under varying resource constraints and training intensities.

**Effect of Local Epochs.** To further assess the robustness of our method under different local training budgets, we increase the number of local epochs to $E = 4$. As shown in Table 6, FedTreeLoRA consistently maintains its superiority. Our tree-structured aggregation effectively mitigates drift by ensuring that clients only aggregate with structurally similar peers, thereby preserving high performance even with increased local steps.

*Table 6.* Performance with increased Local Epochs ($E = 4$).

| Methods | MNLI | QNLI | SST2 | QQP | Average |
|---|---|---|---|---|---|
| FedIT (Zhang et al., 2024) | $82.47 \pm 0.36$ | $86.36 \pm 0.37$ | $93.01 \pm 0.99$ | $83.83 \pm 0.60$ | 86.42 |
| FFA-LoRA ($r = 4$) (Sun et al., 2024) | $82.68 \pm 0.49$ | $90.32 \pm 0.25$ | $93.32 \pm 0.10$ | $83.16 \pm 0.42$ | 87.37 |
| FFA-LoRA ($r = 8$) (Sun et al., 2024) | $82.75 \pm 0.59$ | $90.73 \pm 0.64$ | $93.82 \pm 0.18$ | $83.66 \pm 0.17$ | 87.74 |
| FedSA (Guo et al., 2025) | $84.12 \pm 0.64$ | $92.15 \pm 0.45$ | $96.07 \pm 0.41$ | $89.15 \pm 0.10$ | 90.37 |
| FedDPA (Yang et al., 2024) | $84.69 \pm 0.54$ | $92.20 \pm 0.53$ | $95.91 \pm 0.40$ | $89.50 \pm 0.25$ | 90.58 |
| FedALT (Bian et al., 2026) | $86.28 \pm 0.25$ | $91.70 \pm 0.38$ | $96.51 \pm 0.20$ | $88.97 \pm 0.53$ | 90.87 |
| FedLEASE (Wang et al., 2025) | $86.48 \pm 0.34$ | $93.51 \pm 0.27$ | $96.18 \pm 0.58$ | $90.13 \pm 0.39$ | 91.58 |
| **FedTreeLoRA (Ours)** | $\mathbf{88.27 \pm 0.18}$ | $\mathbf{94.03 \pm 0.14}$ | $\mathbf{96.88 \pm 0.15}$ | $\mathbf{92.07 \pm 0.28}$ | **92.81** |

**Effect of LoRA Rank.** To assess parameter efficiency and capacity, we evaluated performance under a low-rank regime ($r = 2$) and a high-rank regime ($r = 6$). The results are summarized in Table 7 and 8. FedTreeLoRA maintains its lead over baselines in both settings. In the strictly constrained $r = 2$ setting, the efficiency of our layer-wise expert allocation becomes critical, yielding gains over the rigid allocation strategies of methods like FedLEASE.

*Table 7.* Performance comparison under different LoRA Ranks ($r = 2$).

| Methods | MNLI | QNLI | SST2 | QQP | Average |
|---|---|---|---|---|---|
| FedIT (Zhang et al., 2024) | $81.08 \pm 0.46$ | $86.15 \pm 0.58$ | $92.42 \pm 0.50$ | $83.78 \pm 0.46$ | 85.86 |
| FFA-LoRA (Sun et al., 2024) | $81.74 \pm 0.56$ | $86.41 \pm 0.19$ | $93.13 \pm 0.37$ | $83.16 \pm 0.73$ | 86.11 |
| FedSA (Guo et al., 2025) | $82.55 \pm 0.90$ | $90.79 \pm 0.27$ | $95.24 \pm 0.55$ | $89.33 \pm 0.49$ | 89.48 |
| FedDPA (Yang et al., 2024) | $82.74 \pm 0.60$ | $91.07 \pm 0.33$ | $95.16 \pm 0.19$ | $89.69 \pm 0.67$ | 89.67 |
| FedALT (Bian et al., 2026) | $84.45 \pm 0.30$ | $90.90 \pm 0.18$ | $95.83 \pm 0.17$ | $89.49 \pm 0.54$ | 90.17 |
| FedLEASE (Wang et al., 2025) | $85.49 \pm 0.29$ | $92.77 \pm 0.20$ | $95.02 \pm 0.27$ | $90.35 \pm 0.38$ | 90.91 |
| **FedTreeLoRA (Ours)** | $\mathbf{86.60 \pm 0.43}$ | $\mathbf{93.48 \pm 0.16}$ | $\mathbf{96.30 \pm 0.39}$ | $\mathbf{91.46 \pm 0.08}$ | **91.96** |

*Table 8.* Performance comparison under different LoRA Ranks ($r = 6$).

| Methods | MNLI | QNLI | SST2 | QQP | Average |
|---|---|---|---|---|---|
| FedIT (Zhang et al., 2024) | $82.84 \pm 0.24$ | $87.78 \pm 0.52$ | $93.61 \pm 0.52$ | $84.17 \pm 0.29$ | 87.10 |
| FFA-LoRA (Sun et al., 2024) | $82.78 \pm 0.93$ | $88.43 \pm 0.58$ | $93.87 \pm 0.69$ | $83.49 \pm 0.78$ | 87.14 |
| FedSA (Guo et al., 2025) | $83.70 \pm 0.24$ | $91.30 \pm 0.85$ | $96.05 \pm 0.14$ | $89.83 \pm 0.67$ | 90.22 |
| FedDPA (Yang et al., 2024) | $84.03 \pm 0.53$ | $91.20 \pm 0.82$ | $96.02 \pm 0.34$ | $89.45 \pm 0.53$ | 90.18 |
| FedALT (Bian et al., 2026) | $81.45 \pm 0.41$ | $91.93 \pm 0.13$ | $96.12 \pm 0.42$ | $89.50 \pm 0.18$ | 89.75 |
| FedLEASE (Wang et al., 2025) | $86.03 \pm 0.15$ | $93.34 \pm 0.36$ | $95.25 \pm 0.79$ | $90.66 \pm 0.20$ | 91.32 |
| **FedTreeLoRA (Ours)** | $\mathbf{88.37 \pm 0.32}$ | $\mathbf{94.13 \pm 0.22}$ | $\mathbf{96.71 \pm 0.08}$ | $\mathbf{91.45 \pm 0.20}$ | **92.67** |

**Impact of Client Numbers.** We evaluate scalability by varying the system size to $N = 10$ and $N = 40$ while maintaining the same total dataset size to simulate data fragmentation. As shown in Table 9 and 10, FedTreeLoRA demonstrates superior scalability. Flat clustering methods could struggle to form robust clusters due to noise. In contrast, our hierarchical structure allows sparse clients to share knowledge effectively at the root (shallow layers) while specializing only where necessary.

*Table 9.* Performance comparison under different Client Numbers ($N = 10$).

| Methods | MNLI | QNLI | SST2 | QQP | Average |
|---|---|---|---|---|---|
| FedIT (Zhang et al., 2024) | $82.33 \pm 0.50$ | $89.20 \pm 0.35$ | $93.14 \pm 0.21$ | $84.91 \pm 0.57$ | 87.40 |
| FFA-LoRA (r = 4) (Sun et al., 2024) | $82.40 \pm 0.89$ | $89.68 \pm 0.85$ | $93.34 \pm 0.16$ | $84.93 \pm 0.74$ | 87.59 |
| FFA-LoRA (r = 8) (Sun et al., 2024) | $83.33 \pm 0.35$ | $90.10 \pm 0.21$ | $93.60 \pm 0.48$ | $85.43 \pm 0.73$ | 88.12 |
| FedSA (Guo et al., 2025) | $83.52 \pm 0.92$ | $90.78 \pm 0.35$ | $94.75 \pm 0.63$ | $91.35 \pm 0.17$ | 90.10 |
| FedDPA (Yang et al., 2024) | $83.73 \pm 0.30$ | $90.25 \pm 0.47$ | $94.91 \pm 0.23$ | $91.15 \pm 0.13$ | 90.01 |
| FedALT (Bian et al., 2026) | $83.50 \pm 0.71$ | $90.68 \pm 0.60$ | $94.72 \pm 0.49$ | $91.14 \pm 0.16$ | 90.01 |
| FedLEASE (Wang et al., 2025) | $86.19 \pm 0.12$ | $91.97 \pm 0.20$ | $94.82 \pm 0.41$ | $91.45 \pm 0.26$ | 91.11 |
| **FedTreeLoRA (Ours)** | $\mathbf{87.10 \pm 0.21}$ | $\mathbf{92.88 \pm 0.42}$ | $\mathbf{95.76 \pm 0.30}$ | $\mathbf{92.65 \pm 0.36}$ | **92.10** |

*Table 10.* Performance comparison under different Client Numbers ($N = 40$).

| Methods | MNLI | QNLI | SST2 | QQP | Average |
|---|---|---|---|---|---|
| FedIT (Zhang et al., 2024) | $79.28 \pm 0.56$ | $85.39 \pm 0.74$ | $91.99 \pm 0.71$ | $80.05 \pm 0.97$ | 84.18 |
| FFA-LoRA (r = 4) (Sun et al., 2024) | $81.71 \pm 0.46$ | $86.62 \pm 0.79$ | $91.97 \pm 0.74$ | $82.41 \pm 0.72$ | 85.68 |
| FFA-LoRA (r = 8) (Sun et al., 2024) | $82.35 \pm 0.48$ | $86.98 \pm 0.62$ | $92.74 \pm 0.85$ | $82.71 \pm 0.62$ | 86.20 |
| FedSA (Guo et al., 2025) | $83.88 \pm 0.41$ | $89.13 \pm 0.31$ | $94.08 \pm 0.54$ | $85.96 \pm 0.47$ | 88.26 |
| FedDPA (Yang et al., 2024) | $83.45 \pm 0.09$ | $88.53 \pm 0.77$ | $93.52 \pm 0.78$ | $85.63 \pm 0.69$ | 87.78 |
| FedALT (Bian et al., 2026) | $82.53 \pm 0.44$ | $89.11 \pm 0.33$ | $91.78 \pm 0.42$ | $85.68 \pm 0.05$ | 87.28 |
| FedLEASE (Wang et al., 2025) | $84.13 \pm 0.35$ | $90.99 \pm 0.08$ | $94.11 \pm 0.94$ | $87.71 \pm 0.15$ | 89.24 |
| **FedTreeLoRA (Ours)** | $\mathbf{87.28 \pm 0.33}$ | $\mathbf{92.20 \pm 0.30}$ | $\mathbf{94.87 \pm 0.26}$ | $\mathbf{89.48 \pm 0.10}$ | **90.96** |

### C.3. Robustness to Data Distribution Shifts

Finally, we examine the resilience of our method against different forms of statistical heterogeneity, varying from label skew to distinct task assignments.

**Label Distribution Skew.** In the main experiments, we have already evaluated FedTreeLoRA under a highly heterogeneous setting ($\alpha = 0.5$). To further examine robustness to label skew, we additionally consider two milder non-IID regimes with $\alpha \in \{0.7, 1\}$. Both settings introduce label imbalance across clients while being less extreme than the primary benchmark.

Across these configurations, FedTreeLoRA consistently achieves the best performance among all compared methods. This indicates that the proposed tree-structured aggregation remains reliably effective even as the degree of heterogeneity varies, and does not rely on extremely skewed data to demonstrate benefits. As $\alpha$ increases and the client distributions become closer to IID, our adaptive layer-wise specialization mechanism naturally performs fewer splits, promoting more parameter sharing while still preserving the hierarchical structure when beneficial. These results further validate the robustness of FedTreeLoRA to varying levels of label skew, as summarized in Table 11 and Table 12.

**Task Heterogeneity.** Following the protocol of FedLEASE (Wang et al., 2025), we evaluate a task-heterogeneous setting where client diversity arises from performing different NLP tasks rather than label skew. We simulate a federated system with $N = 16$ clients, partitioned into four groups of four label IID clients each, corresponding to four GLUE tasks: MNLI, QNLI, SST-2, and QQP. We use Cosine distance in this setting, as it is more suitable for measuring cross-task differences in low-rank update directions.

*Performance Analysis.* As reported in Table 13, FedTreeLoRA achieves the highest average accuracy under task heterogeneity. While FedLEASE is specifically designed for this scenario, it enforces a flat task-wise separation. In contrast, FedTreeLoRA exploits the hierarchical nature of Transformer representations by allowing clients to share low-level linguistic features while gradually specializing toward task-specific decision boundaries. This hierarchical sharing enables stronger positive

*Table 11.* Performance comparison under different Label Skew ($\alpha = 1$).

| Methods | MNLI | QNLI | SST2 | QQP | Average |
|---|---|---|---|---|---|
| FedIT (Zhang et al., 2024) | $83.13 \pm 0.20$ | $88.04 \pm 0.56$ | $93.67 \pm 0.61$ | $83.88 \pm 0.36$ | 87.18 |
| FFA-LoRA (r = 4) (Sun et al., 2024) | $83.41 \pm 0.83$ | $87.82 \pm 0.99$ | $94.12 \pm 0.33$ | $83.78 \pm 0.53$ | 87.28 |
| FFA-LoRA (r = 8) (Sun et al., 2024) | $83.69 \pm 0.13$ | $88.94 \pm 0.23$ | $94.27 \pm 0.60$ | $84.13 \pm 0.23$ | 87.76 |
| FedSA (Guo et al., 2025) | $85.32 \pm 0.08$ | $89.17 \pm 0.97$ | $94.50 \pm 0.40$ | $87.89 \pm 0.39$ | 89.22 |
| FedDPA (Yang et al., 2024) | $85.38 \pm 0.21$ | $89.93 \pm 0.33$ | $94.74 \pm 0.63$ | $88.07 \pm 0.52$ | 89.53 |
| FedALT (Bian et al., 2026) | $85.85 \pm 0.88$ | $90.10 \pm 0.45$ | $94.47 \pm 0.29$ | $87.95 \pm 0.11$ | 89.59 |
| FedLEASE (Wang et al., 2025) | $86.98 \pm 0.06$ | $91.83 \pm 0.14$ | $95.29 \pm 0.61$ | $88.16 \pm 0.20$ | 90.57 |
| **FedTreeLoRA (Ours)** | $\mathbf{87.43 \pm 0.26}$ | $\mathbf{93.03 \pm 0.32}$ | $\mathbf{95.62 \pm 0.31}$ | $\mathbf{89.23 \pm 0.10}$ | **91.33** |

*Table 12.* Performance comparison under different Label Skew ($\alpha = 0.7$).

| Methods | MNLI | QNLI | SST2 | QQP | Average |
|---|---|---|---|---|---|
| FedIT (Zhang et al., 2024) | $81.90 \pm 0.15$ | $87.28 \pm 0.29$ | $93.99 \pm 0.10$ | $82.87 \pm 0.60$ | 86.51 |
| FFA-LoRA (r = 4) (Sun et al., 2024) | $83.11 \pm 0.31$ | $89.18 \pm 0.76$ | $94.30 \pm 0.93$ | $83.23 \pm 0.79$ | 87.46 |
| FFA-LoRA (r = 8) (Sun et al., 2024) | $83.32 \pm 0.93$ | $89.60 \pm 0.27$ | $94.56 \pm 0.36$ | $83.51 \pm 0.56$ | 87.75 |
| FedSA (Guo et al., 2025) | $84.73 \pm 0.39$ | $89.84 \pm 0.22$ | $94.68 \pm 0.39$ | $88.98 \pm 0.27$ | 89.56 |
| FedDPA (Yang et al., 2024) | $84.19 \pm 0.36$ | $89.69 \pm 0.39$ | $94.79 \pm 0.51$ | $89.13 \pm 0.39$ | 89.45 |
| FedALT (Bian et al., 2026) | $83.23 \pm 0.87$ | $89.55 \pm 0.11$ | $90.86 \pm 0.51$ | $88.71 \pm 0.19$ | 88.09 |
| FedLEASE (Wang et al., 2025) | $85.80 \pm 0.33$ | $92.70 \pm 0.13$ | $95.65 \pm 0.09$ | $90.23 \pm 0.14$ | 91.10 |
| **FedTreeLoRA (Ours)** | $\mathbf{87.90 \pm 0.13}$ | $\mathbf{93.12 \pm 0.10}$ | $\mathbf{96.40 \pm 0.32}$ | $\mathbf{91.31 \pm 0.01}$ | **92.18** |

transfer in shallow layers while preventing negative interference in deeper layers.

*Structural Analysis.* Figure 7 visualizes the learned layer-wise clustering structure. We observe a clear and interpretable hierarchy that aligns with task semantics. All 16 clients initially share a single cluster in shallow layers, reflecting common syntactic and lexical representations. At intermediate depths, the model separates MNLI and QQP clients into distinct branches, while QNLI and SST-2 remain grouped, indicating that their low-rank updates remain geometrically similar at this stage. In deeper layers, the hierarchy further refines into four task-specific clusters, each corresponding to one GLUE task.

This progression from a global cluster to intermediate task groups and finally to task-specific leaves demonstrates that FedTreeLoRA does not impose a rigid task partition. Instead, it automatically discovers a hierarchical task topology that reflects both shared linguistic structure and task-dependent specialization. Such adaptive, depth-aware decomposition is precisely what enables FedTreeLoRA to outperform flat task separation methods in heterogeneous federated environments.

*Table 13.* Performance under Task Heterogeneity ($N = 16$ clients, 4 per task).

| Methods | MNLI | QNLI | SST2 | QQP | Average |
|---|---|---|---|---|---|
| FedIT (Zhang et al., 2024) | $71.46 \pm 1.74$ | $83.71 \pm 1.95$ | $93.25 \pm 0.31$ | $78.54 \pm 3.03$ | 81.74 |
| FFA-LoRA (Sun et al., 2024) | $72.17 \pm 2.74$ | $82.75 \pm 2.26$ | $93.79 \pm 0.68$ | $81.25 \pm 2.56$ | 82.49 |
| FedSA (Guo et al., 2025) | $74.29 \pm 1.09$ | $86.33 \pm 0.75$ | $93.87 \pm 0.18$ | $82.12 \pm 0.74$ | 84.15 |
| FedDPA (Yang et al., 2024) | $75.00 \pm 1.70$ | $86.29 \pm 0.16$ | $93.58 \pm 0.16$ | $81.92 \pm 2.01$ | 84.20 |
| FedALT (Bian et al., 2026) | $73.25 \pm 1.40$ | $83.00 \pm 1.35$ | $93.69 \pm 0.19$ | $84.31 \pm 0.44$ | 83.56 |
| FedLEASE (Wang et al., 2025) | $79.91 \pm 1.86$ | $87.54 \pm 1.35$ | $94.17 \pm 0.37$ | $83.46 \pm 0.79$ | 86.27 |
| **FedTreeLoRA (Ours)** | $\mathbf{82.94 \pm 0.94}$ | $\mathbf{89.31 \pm 0.19}$ | $\mathbf{94.19 \pm 0.31}$ | $\mathbf{84.75 \pm 1.25}$ | **87.80** |

## C.4. Impact of Different Distance Metrics

To evaluate the sensitivity of FedTreeLoRA to the geometric definition of client similarity, we conduct a comparative study using two widely adopted distance functions for constructing the global topological skeleton: *Frobenius distance* (our default) and *Cosine distance*. The Frobenius distance captures the absolute magnitude of parameter deviations, while Cosine

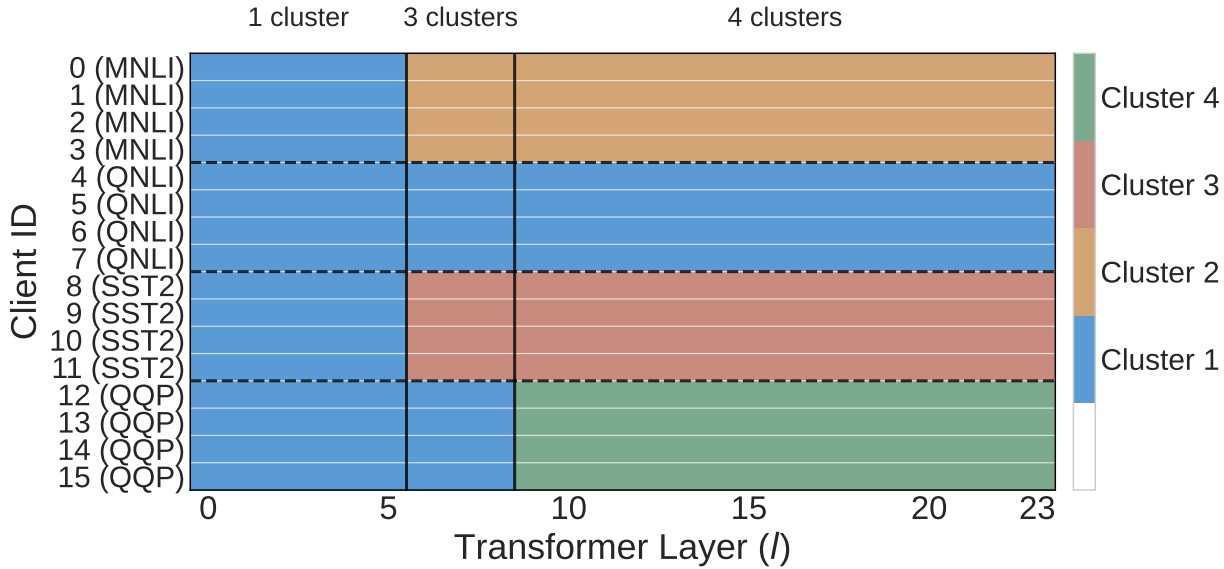

*Figure 7.* Layer-wise client clustering under task heterogeneity. Clients assigned to the same GLUE task form stable branches at deeper layers, while all tasks share common representations in shallow layers, illustrating the hierarchical specialization discovered by FedTreeLoRA.

distance focuses on the directional orientation of the LoRA updates.

As summarized in Table 14, the performance remains remarkably consistent across both metrics. On the GLUE benchmark, the average accuracy gap between the two implementations is a marginal 0.09% (92.45% for Cosine vs. 92.36% for Frobenius). The consistency underscores the robustness of FedTreeLoRA, demonstrating that its performance gains are rooted in the **structural hierarchy** and **layer-wise alignment** rather than the specific distance definition. We default to Frobenius distance for label non-IID but task-homogeneous settings, where LoRA update differences are mainly reflected in their magnitudes.

*Table 14.* Sensitivity analysis of distance metrics on GLUE benchmarks. Results demonstrate that `FedTreeLoRA` is metric-agnostic, maintaining high performance regardless of the specific similarity definition.

| Metric | MNLI | QNLI | SST2 | QQP | Average |
|---|---|---|---|---|---|
| Cosine Distance | $88.29 \pm 0.30$ | $93.47 \pm 0.25$ | $96.41 \pm 0.02$ | $91.61 \pm 0.09$ | **92.45** |
| Frobenius Distance | $88.15 \pm 0.25$ | $93.37 \pm 0.62$ | $96.56 \pm 0.07$ | $91.35 \pm 0.17$ | 92.36 |

**C.5. Effect of Warm-up Duration and Fixed Tree Topology**

In FedTreeLoRA, the tree topology is constructed once after the warm-up stage and then kept fixed during subsequent optimization. The rationale behind this design is that the warm-up phase already provides a sufficiently informative estimate of inter-client relationships, rather than merely serving as a rough initialization. As illustrated in Figure 7, the learned hierarchy exhibits strong semantic consistency: clients tend to remain unified in shallow layers while progressively separating into task-related subgroups at deeper layers. This observation suggests that the discovered topology captures a stable and interpretable collaborative structure.

To further validate whether a short warm-up is sufficient for reliable topology discovery, we vary the number of warm-up epochs and report the downstream performance in Table 15. The results show that FedTreeLoRA remains highly stable across different warm-up durations. Increasing the warm-up stage from 5 to 10 epochs yields nearly identical average accuracy, indicating that the hierarchical relationships among clients emerge early during adaptation.

Based on these observations, FedTreeLoRA decouples topology discovery from parameter optimization. Maintaining a fixed hierarchy throughout training preserves structural consistency across layers while avoiding repeated similarity computation

*Table 15.* Sensitivity analysis of warm-up duration on GLUE benchmarks. Results show that FedTreeLoRA remains highly stable across different warm-up stages, indicating that reliable client topology can be identified early during adaptation.

| Warm-up Epochs | MNLI | QNLI | SST2 | QQP | Average |
|---|---|---|---|---|---|
| 5 | 82.94 | 89.31 | 94.19 | 84.75 | 87.80 |
| 10 | 82.66 | 89.83 | 94.62 | 84.13 | 87.81 |

and hierarchical clustering at every communication round, thereby significantly reducing computational overhead.

### C.6. Robustness to Dynamic Client Participation

Although FedTreeLoRA primarily focuses on personalized federated fine-tuning under a fixed client population, an important practical scenario is dynamic client participation, where new clients may join the system during training. While this setting is more closely related to federated continual learning, FedTreeLoRA can be naturally extended to support incremental client integration. Specifically, when a new client joins, the client first performs a short local warm-up stage to obtain an initial LoRA representation. The resulting LoRA $B$ matrices are then used to measure similarity against the existing hierarchy. Based on this similarity, the client is assigned to the closest branch of the learned tree and subsequently follows the same layer-wise aggregation strategy as existing clients.

To evaluate this behavior, we conduct experiments on the MNLI benchmark where a new client joins the training at communication round 10 or 20. For each setting, we randomly sample a different client to join the system in order to avoid bias toward a specific client distribution. We report: (1) the performance of the original clients without client joining, (2) the performance of existing clients after the new client joins, (3) the performance of the new client after joining the hierarchy, and (4) a local-only baseline where the new client trains independently without collaboration.

*Table 16.* Robustness to dynamic client participation on MNLI. Results show that integrating new clients causes negligible degradation to existing clients while significantly improving the performance of the newly joined client compared to local-only training.

| Join Round | Original Clients | Existing Clients | New Client | Local Only |
|---|---|---|---|---|
| 10 | 87.90 | 87.43 | 84.50 | 82.00 |
| 20 | 87.90 | 87.50 | 86.50 | 79.00 |

The results demonstrate that the learned hierarchy provides a stable and reusable topology for incremental personalization. Existing clients experience only marginal performance changes after client insertion, while the newly joined client consistently benefits from collaborative adaptation compared to isolated local training.

## D. Detailed Results of Ablation Studies

Due to space limitations in the main paper, we reported only the average accuracy (Avg. Acc.) to summarize the findings of our ablation studies. To provide a more comprehensive and granular analysis of how each component of FedTreeLoRA contributes to the results, we here report the detailed performance breakdown for each of the four evaluation tasks: MNLI, QNLI, SST2, and QQP, corresponding to the summarized results presented in the main paper.

### D.1. Detailed Analysis of Interaction Mechanisms

As discussed in the main paper, we compared our *Cluster-External* aggregation against three variants: Isolationist, Decomposed Experts, and the MoE Router. Table 17 provides the task-specific accuracy for each mechanism, providing insight into the trade-off between performance and efficiency.

### D.2. Detailed Impact of Layer-wise Adaptivity

Table 18 extends the results from the main paper regarding layer-wise adaptivity. It compares our adaptive depth search against fixed clustering granularities ($k = 1, 4, 8$) across each specific task to validate the necessity of layer-wise adaptivity.

*Table 17.* Detailed performance of different Interaction Mechanisms across GLUE tasks. Accuracy is reported in (%).

| Interaction Variant | MNLI | QNLI | SST2 | QQP | Average |
|---|---|---|---|---|---|
| Isolationist (Cluster-Only) | $86.60 \pm 0.21$ | $92.57 \pm 0.44$ | $95.96 \pm 0.39$ | $90.48 \pm 0.39$ | 91.40 |
| Decomposed Experts | $\mathbf{88.46 \pm 0.23}$ | $\mathbf{93.91 \pm 0.36}$ | $96.54 \pm 0.04$ | $\mathbf{91.35 \pm 0.13}$ | **92.57** |
| MoE Router | $87.92 \pm 0.82$ | $93.35 \pm 0.58$ | $96.15 \pm 0.46$ | $90.67 \pm 0.45$ | 92.02 |
| **VAScalar-Mixed** | $88.15 \pm 0.25$ | $93.37 \pm 0.62$ | $\mathbf{96.56 \pm 0.07}$ | $\mathbf{91.35 \pm 0.17}$ | 92.36 |

*Table 18.* Detailed performance of Fixed-Depth vs. Layer-wise Adaptive aggregation.

| Aggregation Depth | MNLI | QNLI | SST2 | QQP | Average |
|---|---|---|---|---|---|
| Fixed ($k=1$, Global Only) | $83.18 \pm 0.74$ | $87.03 \pm 0.43$ | $93.65 \pm 0.63$ | $84.93 \pm 0.59$ | 87.20 |
| Fixed ($k=4$, Coarse) | $86.75 \pm 0.63$ | $92.95 \pm 0.22$ | $96.13 \pm 0.16$ | $89.98 \pm 0.59$ | 91.45 |
| Fixed ($k=8$, Fine-grained) | $85.81 \pm 0.68$ | $92.02 \pm 0.71$ | $95.17 \pm 0.13$ | $89.96 \pm 0.89$ | 90.74 |
| **Layer-wise Adaptive ($c_l^*$)** | $\mathbf{88.15 \pm 0.25}$ | $\mathbf{93.37 \pm 0.62}$ | $\mathbf{96.56 \pm 0.07}$ | $\mathbf{91.35 \pm 0.17}$ | **92.36** |

### D.3. Effect of the Global Tree Skeleton

Finally, we report the detailed breakdown of the structural consistency analysis presented in the main paper. Table 19 demonstrates the importance of the global tree skeleton in maintaining topological coherence across model layers for each individual dataset.

*Table 19.* Detailed impact of Global Structural Consistency on individual datasets.

| Structural Prior | MNLI | QNLI | SST2 | QQP | Average |
|---|---|---|---|---|---|
| Independent Clustering | $83.84 \pm 0.32$ | $89.74 \pm 1.95$ | $95.06 \pm 0.21$ | $89.23 \pm 1.56$ | 89.47 |
| **Global Tree Skeleton** | $\mathbf{88.15 \pm 0.25}$ | $\mathbf{93.37 \pm 0.62}$ | $\mathbf{96.56 \pm 0.07}$ | $\mathbf{91.35 \pm 0.17}$ | **92.36** |

## E. Computational Overhead and Wall-Clock Training Time

We analyze the computational efficiency of FedTreeLoRA compared to baseline methods.

**Fairness of Training Budget.** To ensure a fair comparison, we maintain an identical total training budget across all evaluated methods. For NLU tasks, the total communication budget is fixed at $T = 30$ rounds. For methods requiring a warmup phase to capture client relationships (e.g. FedLEASE and the proposed FedTreeLoRA), we allocate $E_{warm} = 10$ epochs (i.e. 5 rounds) for local warmup, followed by 25 rounds of federated fine-tuning. Similarly, for NLG tasks, the total budget is $T = 10$ rounds, with $E_{warm} = 2$ epochs (i.e. 1 round) dedicated to the warmup phase. This configuration ensures that all methods undergo the same cumulative number of local updates and communication iterations, isolating the performance gains to the structural aggregation mechanism rather than an increased training budget.

**Efficiency of Topology Generation.** Our tree topology generation step is performed **only once** during the initialization phase (immediately following warmup) and is not repeated during the iterative communication rounds. Consequently, its impact on the total training time is negligible. Furthermore, utilizing only the LoRA $B$ matrices for similarity computation provides an efficient and lightweight proxy for task alignment. This is because the dimensionality of $B$ is significantly smaller than that of full model weights or accumulated $BA$ products.

As shown in Table 20, we measured the topology generation time to be approximately 4.21 seconds on an Intel Xeon Platinum 8570 CPU. This is orders of magnitude shorter than the total training duration, which is approximately 70.82 seconds when utilizing NVIDIA B200 GPUs. Notably, FedTreeLoRA achieves superior performance with a total training time comparable to standard FedAvg-based methods like FedIT (65.15s), and remains lower than complex clustering-based baselines such as FedLEASE (77.99s) which require heavy routing computations and additional parameter overhead.

*Table 20.* Comparison of Computational Overhead. All times are measured in seconds (s).

| Time (s) | FedIT | FFA-LoRA | FedSA | FedDPA | FedALT | FedLEASE | FedTreeLoRA |
|---|---|---|---|---|---|---|---|
| Local Training (Per Epoch) | 1.0545 | 1.0369 | 1.0582 | 1.0668 | 1.2091 | 1.1883 | **1.0716** |
| Global Aggregation | 0.0628 | 0.0594 | 0.0693 | 0.0723 | 0.0782 | 0.0908 | **0.0770** |
| Topology/Clustering Time | - | - | - | - | - | 3.97 | **4.21** |
| **Total Training Time** | **65.154** | **63.996** | **65.571** | **66.177** | **74.892** | **77.992** | **70.816** |

## F. Extended NLG Results on an Alternative LLM Family

To evaluate the architectural robustness of **FedTreeLoRA**, we further extend our Natural Language Generation (NLG) experiments by instantiating the framework on a large language model from an alternative LLM family, specifically **BLOOM-7B** (Le Scao et al., 2023).

**Experimental Configuration.** Following the protocol established in Section 6.2, we utilize $N = 8$ clients partitioned into four distinct NLG task groups from the FLAN collection: *Text Editing*, *Struct to Text*, *Sentiment Analysis*, and *Commonsense Reasoning*. Each task is assigned to two clients to simulate task-level heterogeneity. We maintain identical hyperparameters to the main LLaMA-2 experiments: a batch size of 8, $E = 2$ local epochs, and $T = 10$ communication rounds using the AdamW optimizer. LoRA adapters with rank $r = 8$ are applied to the query and value projections. Consistent with standard evaluation protocols, we report the average ROUGE-1 scores.

**Results and Analysis.** As shown in Table 21, **FedTreeLoRA** consistently outperforms both general federated fine-tuning baselines and flat-clustering methods on the BLOOM model. The results indicate that the "Flat-Model Assumption", which treats LLM layers as a monolithic block for aggregation, is equally restrictive for BLOOM. By dynamically constructing an aggregation tree that aligns with the model's functional depth, our method mitigates negative transfer between conflicting domains while maximizing the utility of shared linguistic knowledge.

*Table 21.* Performance comparison on NLG benchmarks using BLOOM. We report ROUGE-1 scores averaged over three independent runs.

| Methods | Text Edit | Struct2Text | Sentiment | Reasoning | Average |
|---|---|---|---|---|---|
| FedIT (Zhang et al., 2024) | $77.21 \pm 1.39$ | $51.80 \pm 0.89$ | $49.73 \pm 0.36$ | $72.37 \pm 0.58$ | 62.78 |
| FFA-LoRA (r = 8) (Sun et al., 2024) | $77.05 \pm 1.35$ | $51.10 \pm 0.66$ | $48.72 \pm 1.49$ | $72.32 \pm 0.46$ | 62.30 |
| FFA-LoRA (r = 16) (Sun et al., 2024) | $78.77 \pm 1.81$ | $51.20 \pm 0.60$ | $49.05 \pm 0.84$ | $72.12 \pm 0.86$ | 62.79 |
| FedSA (Guo et al., 2025) | $84.93 \pm 1.34$ | $52.93 \pm 0.23$ | $49.38 \pm 0.84$ | $72.64 \pm 1.18$ | 64.94 |
| FedDPA (Yang et al., 2024) | $88.56 \pm 0.86$ | $53.78 \pm 0.36$ | $51.31 \pm 0.51$ | $72.16 \pm 1.78$ | 66.45 |
| FedALT (Bian et al., 2026) | $88.32 \pm 0.19$ | $53.58 \pm 0.19$ | $50.97 \pm 1.13$ | $73.22 \pm 1.45$ | 66.52 |
| FedLEASE (Wang et al., 2025) | $87.07 \pm 0.45$ | $52.22 \pm 0.39$ | $52.12 \pm 0.15$ | $73.75 \pm 0.90$ | 66.79 |
| **FedTreeLoRA (Ours)** | $\mathbf{88.84 \pm 0.34}$ | $\mathbf{55.20 \pm 1.01}$ | $\mathbf{52.85 \pm 0.45}$ | $\mathbf{74.23 \pm 0.68}$ | **67.78** |

## G. Justification for Similarity Measurement via LoRA $B$ Matrices

In FedTreeLoRA, we utilize the distance between clients' LoRA $B$ matrices to construct the topological tree. This design choice is grounded in the asymmetric roles of adapter matrices observed in recent federated fine-tuning literature (Guo et al., 2025; Wang et al., 2025), which suggest that $A$ matrices tend to capture general features while $B$ matrices encode task-specific semantics. To verify and refine this hypothesis in a layer-wise and heterogeneity-aware manner, we conduct a controlled analysis on RoBERTa-Large for the QNLI task with two clients under an IID setting (identical label distributions) and a highly non-IID setting ($[0.8, 0.2]$ vs. $[0.2, 0.8]$), reporting the average pairwise Frobenius distance of LoRA parameters across all 24 Transformer layers. As shown in Figure 8, the upper row (Proj $A$ @ Query and Value) exhibits nearly identical distance trajectories under IID and non-IID across all layers, indicating that $A$ learns a shared low-rank basis that is largely insensitive to data heterogeneity. In contrast, the lower row (Proj $B$ @ Query and Value) consistently separates IID and non-IID settings, and this separation is strongly depth-dependent: in shallow layers the distances are close, reflecting shared general representations, whereas in deeper layers the non-IID distance grows substantially while the IID distance remains small, producing a widening gap that directly reflects the emergence of client-specific semantics. This layer-wise divergence

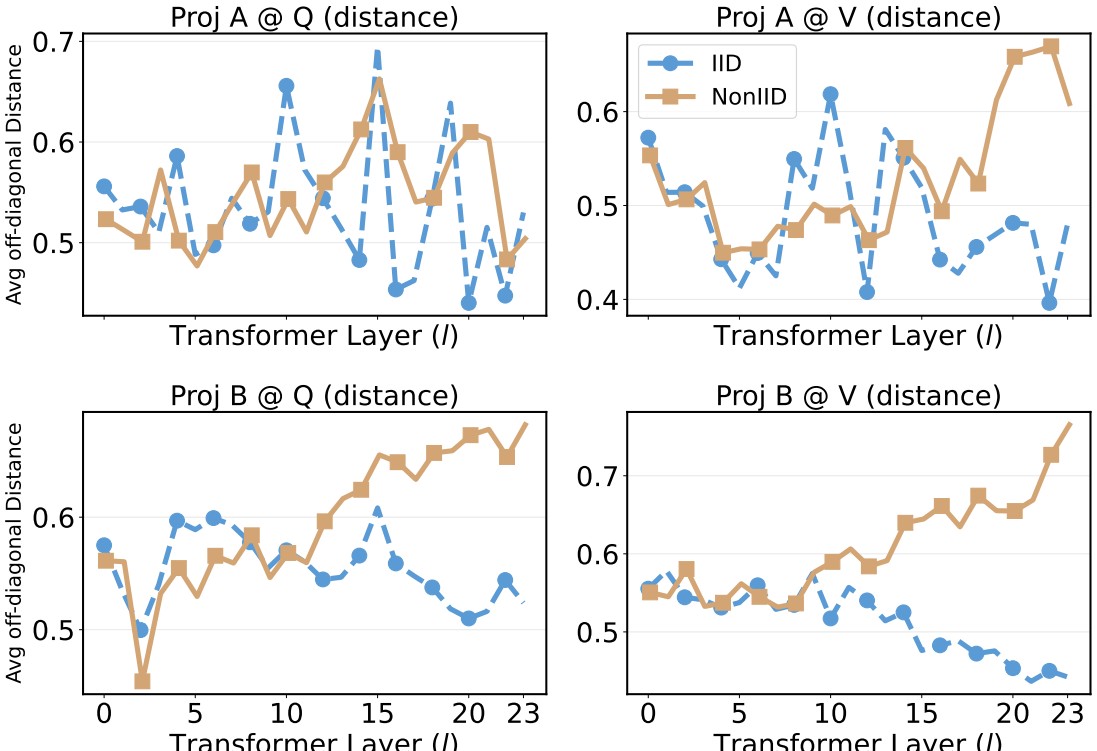

*Figure 8.* Layer-wise average pairwise Frobenius distance between two clients on QNLI under IID (identical label distributions) and Non-IID ($[0.8, 0.2]$ vs. $[0.2, 0.8]$) settings. The upper-left and upper-right panels (Proj $A$ at Query and Value) show nearly overlapping IID and Non-IID curves across all layers, indicating that $A$ captures distribution-invariant shared representations. The lower-left and lower-right panels (Proj $B$ at Query and Value) exhibit a progressively widening gap between IID and Non-IID as depth increases, revealing that client-specific semantics emerge primarily in deeper layers through the LoRA $B$ matrices. This validates using $B$-matrix distances to quantify heterogeneity and to guide hierarchical, layer-wise aggregation in FedTreeLoRA.

in $B$ explains why shallow layers can be safely aggregated while deeper layers require personalized or clustered aggregation, and it establishes LoRA $B$ matrices as a sensitive and reliable signal for measuring client similarity. Since computing distances on $B \in \mathbb{R}^{d_{\text{out}} \times r}$ is orders of magnitude cheaper than on the full adapter product $BA \in \mathbb{R}^{d_{\text{out}} \times d_{\text{in}}}$, using $B$ to construct the topological tree is both empirically justified and computationally efficient.

## H. Compared Methods

We provide a detailed overview of the baseline methods used in our evaluation, highlighting their operational mechanisms and distinguishing them from our proposed **FedTreeLoRA**.

**FedIT [ICASSP 2024] (Zhang et al., 2024).** FedIT represents the canonical application of Federated Learning to Instruction Tuning. It directly integrates LoRA with the standard FedAvg framework, aggregating all client updates globally. Under heterogeneous settings, this leads to significant "negative transfer" or cross-client interference, where the global aggregate dilutes task-specific features, often resulting in performance inferior to local-only training.

**FFA-LoRA [ICLR 2024] (Sun et al., 2024).** FFA-LoRA focuses on communication efficiency by freezing the randomly initialized non-zero $A$ matrices and fine-tuning only the zero-initialized $B$ matrices. While this strategy reduces communication overhead, it severely constrains the expressivity of the adaptation. By locking the input projection $A$, the model loses the flexibility to align diverse client inputs into a shared latent space, resulting in suboptimal performance.

**FedSA [ICLR 2025] (Guo et al., 2025).** FedSA proposes a selective aggregation strategy to address heterogeneity. It decomposes the LoRA update into global and local components: only the $A$ matrices are aggregated server-side to learn shared representations, while $B$ matrices remain strictly local to preserve personalization. While FedSA attempts to decouple personalization from generalization, it relies on a rigid, manually defined split. It overlooks that client-specific $B$ matrices

often contain valuable transferable knowledge for distinct sub-communities of clients, which FedTreeLoRA captures via its hierarchical aggregation.

**FedDPA [NeurIPS 2024] (Yang et al., 2024).** FedDPA targets task heterogeneity by employing a dual-adapter architecture. It maintains both a global LoRA module (aggregated via FedAvg) and a local LoRA module (kept private). It utilizes a multi-phase training strategy to balance general knowledge acquisition and local refinement. FedDPA fundamentally relies on a "Global vs. Local" binary separation. It lacks the granularity to model relationships, where groups of clients share specific tasks. FedTreeLoRA replaces this binary dichotomy with a continuous tree structure, allowing parameter sharing to adapt dynamically based on cluster similarity rather than just global or local isolation.

**FedALT [AAAI 2026] (Bian et al., 2026).** FedALT introduces a "Rest-of-World" (RoW) paradigm. Instead of a global model, each client maintains an *Individual LoRA* (trained locally) and a *RoW LoRA* (a frozen aggregate of all other clients' modules). An input-dependent adaptive mixer (resembling a gate) dynamically weights the contribution of the individual and RoW modules during inference. FedALT mitigates interference by isolating the global signal into a frozen module. However, it operates on a coarse-grained "Self vs. Others" logic. It aggregates *all* other clients into a single RoW representation, potentially blending conflicting signals from unrelated tasks. In contrast, FedTreeLoRA selectively aggregates only relevant peers defined by the topological tree, ensuring that shared signals remain semantically coherent.

**FedLEASE [NeurIPS 2025] (Wang et al., 2025).** FedLEASE addresses heterogeneity via flat clustering. It computes client similarity based on initial LoRA updates and groups clients into disjoint clusters, training distinct domain experts for each group. It further employs an adaptive top-$M$ routing mechanism to allow clients to select multiple experts during inference. FedLEASE represents a strong clustering baseline. However, it relies on the *Flat-Model Assumption*: it generates a single partition of clients and applies expert selection uniformly. Crucially, it ignores *vertical heterogeneity*, the fact that shallow layers require broad consensus while deep layers need specialization. FedTreeLoRA advances beyond FedLEASE by constructing a hierarchical tree that enables *layer-wise* depth alignment, allowing the aggregation boundary to shift dynamically from the root (global) to the leaves (personalized) across the model depth.

# I. Clarification with Existing Methods

**Distinction from TreeLoRA (Qian et al., 2025).**  We note that although our method shares the term 'TreeLoRA' with the recent work of (Qian et al., 2025), the two approaches differ fundamentally in problem setting, tree semantics, and optimization role. TreeLoRA (Qian et al., 2025) is proposed for continual learning, where the tree dynamically organizes tasks arriving sequentially based on gradient-direction similarity, and is incrementally expanded via a bandit-based search to mitigate catastrophic forgetting. In contrast, our proposed FedTreeLoRA operates in a federated fine-tuning setting, where the tree models statistical relationships among clients rather than tasks. The tree in FedTreeLoRA is constructed once after a warmup phase using the geometric similarity of LoRA adaptation parameters, serving as a fixed global topological skeleton shared across all Transformer layers. Moreover, the role of the tree is different: in TreeLoRA, the tree directly controls sparse gradient updates by selecting task-specific adapters, whereas in FedTreeLoRA, the tree only defines the topology of parameter sharing, constraining the feasible layer-wise partitions while leaving the parameter interaction mechanism to a Cluster-External expert aggregation with a learnable mixing coefficient. Finally, FedTreeLoRA introduces a layer-wise monotonic depth alignment to ensure structural coherence across model layers, a design that is specific to federated adaptation of deep architectures and has no counterpart in continual learning. Consequently, despite the terminological overlap, FedTreeLoRA represents a distinct federated learning framework focused on structural alignment rather than a variant of tree-based continual learning.

**Additional Distinction from Layer-aware Federated LoRA.**  We further note that recent federated LoRA methods have explored layer-aware adaptation under heterogeneous client resources (Zhang et al., 2025b). However, these approaches primarily focus on resource-efficient layer allocation across clients, rather than modeling the hierarchical relationship between client similarity and aggregation depth. In contrast, FedTreeLoRA studies how parameter sharing should evolve progressively across Transformer layers based on statistical similarity, leading to a tree-structured aggregation topology that jointly reconciles horizontal and vertical heterogeneity.

## J. Reproducibility and Code Availability

To ensure full reproducibility of our experimental results, we publicly release the complete implementation of FedTreeLoRA at: https://github.com/jmbian/FedTreeLoRA.

The repository includes all components required to reproduce the results reported in both the main paper and appendix, including data preprocessing, model architectures, training scripts, evaluation pipelines, global tree construction, layer-wise adaptive clustering, and the cluster-external expert mechanism. We further provide detailed instructions for environment setup, dataset preparation, and command-line execution for all NLU and NLG benchmarks evaluated in this work. In addition, the codebase is designed to support extensibility and ablation studies. Researchers can easily modify the similarity metric (e.g., Frobenius or cosine distance), warm-up duration, search range $K$, and heterogeneity threshold $\tau$, enabling further investigation into the trade-offs between personalization, communication efficiency, and federated adaptation performance.

## K. FedTreeLoRA Algorithm

We provide the completed pseudocode for FedTreeLoRA in Algorithm 1.

---

**Algorithm 1** FedTreeLoRA: Tree-Structured Aggregation

---

**Require:** Clients $\{1, \ldots, N\}$ with datasets $\{\mathcal{D}_k\}$; layers $l = 1, \ldots, L$; warmup epochs $E_{warm}$; rounds $T$; local epochs $E$; search range $K$; threshold $\tau$. $\{\overline{A}_{l,k}^{\text{clus}}, \overline{B}_{l,k}^{\text{clus}}, \lambda_{l,k}\}$.

1: **— Phase 1: Warm-up and Topology Generation —**
2: **Client Side (Parallel):**
3: **for all** $k \in \{1, \ldots, N\}$ **do**
4:     Initialize $\{A_{l,k}, B_{l,k}\}_{l=1}^{L}$;
5:     Warmup-train on $\mathcal{D}_k$ for $E_{warm}$ epochs;
6:     Upload updated $B_{l,k}$ to Server.
7: **end for**
8: **Server Side:**
9: Compute global distance $D_{i,j}^{global} = \frac{1}{L} \sum_{l=1}^{L} \text{dist}(B_{l,i}, B_{l,j})$ via Eq. (3);
10: Build merge tree $\mathcal{T}$ via AHC;
11: Initialize $c_0^* \leftarrow 1$;
12: **for** $l = 1$ **to** $L$ **do**
13:     Compute $D_{i,j}^{(l)} = \text{dist}(B_{l,i}, B_{l,j})$;
14:     Define search space $\Omega_l = \{c \in \mathbb{Z} \mid c_{l-1}^* \leq c \leq \min(N, c_{l-1}^* + K - 1)\}$;
15:     Determine optimal clusters $c_l^* \leftarrow \text{argmax}_{c \in \Omega_l} \phi(c; D^{(l)})$;
16:     Extract partition $P_{c_l^*}$ and assign peer groups for each client.
17: **end for**

18: **— Phase 2: Federated Fine-Tuning —**
19: **for** $t = 1$ **to** $T$ **do**
20:     **Server Side (Aggregation):**
21:     **for all** $k \in \{1, \ldots, N\}$ **do**
22:         **for** $l = 1$ **to** $L$ **do**
23:             Identify peer group $\mathcal{S}_k^{(l)}$ and external group $\mathcal{R}_k^{(l)}$ from $P_{c_l^*}$;
24:             **for all** $\Phi \in \{A, B\}$ **do**
25:                 $\overline{\Phi}_{l,k}^{\text{clus}} \leftarrow \frac{1}{|\mathcal{S}_k^{(l)}|} \sum_{j \in \mathcal{S}_k^{(l)}} \Phi_{l,j}^{(t)}$;
26:                 **if** $|\mathcal{R}_k^{(l)}| > 0$ **then**
27:                     $\overline{\Phi}_{l,k}^{\text{ext}} \leftarrow \frac{1}{|\mathcal{R}_k^{(l)}|} \sum_{j \in \mathcal{R}_k^{(l)}} \Phi_{l,j}^{(t)}$;
28:                 **else**
29:                     $\overline{\Phi}_{l,k}^{\text{ext}} \leftarrow \mathbf{0}$;
30:                 **end if**
31:             **end for**
32:         **end for**
33:         Send aggregated experts $\{\overline{A}_k^{\text{clus/ext}}, \overline{B}_k^{\text{clus/ext}}\}$ to Client $k$.
34:     **end for**
35:     **Client Side (Parallel Update):**
36:     **for all** $k \in \{1, \ldots, N\}$ **do**
37:         Receive experts;
38:         Update $(\overline{A}_{l,k}^{\text{clus}}, \overline{B}_{l,k}^{\text{clus}}, \lambda_{l,k})$ on $\mathcal{D}_k$ for $E$ epochs (via Eq. 8);
39:         Upload updated parameters to Server.
40:     **end for**
41: **end for**

---

