# OpenReview forum: "FedTreeLoRA: Reconciling Statistical and Functional Heterogeneity in Federated LoRA Fine-Tuning"
_ICML.cc/2026/Conference — ICML 2026 regular_

### Official Review · Reviewer_TqYS · 2026-03-02

**Soundness:** 3
**Presentation:** 3
**Significance:** 2
**Originality:** 2
**Overall Recommendation:** 4
**Confidence:** 3

**Summary:**

This paper identifies a critical limitation in existing personalized federated learning (FL) methods using LoRA for LLM fine-tuning: prior works mainly focus on client-side statistical (horizontal) heterogeneity and treat the model as a monolithic block, yet overlook functional (vertical) heterogeneity across LLM layers. To address it, the authors propose a novel LLM fine-tuning in FL with LoRA: FedTreeLoRA. Firstly, they argue that horizontal and vertical heterogeneity are orthogonal but coupled in determining the optimal sharing depth. Then, based on the observation, the paper proposes FedTreeLoRA, which constructs a tree-structured, layer-wise aggregation based on client similarity through three main components: Global Topological Structure Modeling, Adaptive Layer-wise Depth Alignment, and Cluster-External Expert Mechanism.  Furthermore, theoretical convergence analysis is provided, and experiments on NLU and NLG benchmarks demonstrate improvements in personalization and generalization.

**Compliance With Llm Reviewing Policy:**

Affirmed.

**Final Justification:**

The authors’ responses have adequately addressed my concerns on motivation, references, computation overhead, personalization & generalization, and ablation study. Therefore, I will retain my "Weak Accept" recommendation.

**Key Questions For Authors:**

1) L88-90 (left half) states that prior works use coarse-grained client clustering, but L34-42 (right half) mentions that most prior works employ local-global LoRA modules, and only FedLEASE uses client clustering. Could the authors clarify this?
2) Where is the explicit reconciling between generalization and personalization mentioned in L105-108 (left half)? The paper proved personalization through NLU and generalization through NLG. Could the authors clarify the reconciling (is it a trade-off?) between personalization and generalization, rather than improving them independently on two tasks?
3) Can the significance of tree-structured alignment mentioned in section 5.2 be confirmed only through the comparison (Table 3) of the four different mechanisms mentioned in section 5.3 in the ablation experiment? Is it necessary to perform additional ablation on tree-structured alignment to demonstrate its importance?

**Limitations:**

1) FedTreeLoRA demonstrates its strengths in NLU and NLG. However, it remains unclear whether the mentioned observation and conclusions hold consistently in multimodal benchmarks. For example, pFedMMA [1] observed that the sharing of deeper layers can improve model performance more efficiently. This contradiction may cause a negative impact on the application of tree-structured aggregation in multimodal benchmarks.
2) If the client joins and exits the federated system, the topological structure needs to be maintained in real time, further causing the entire training to restart and increasing the computational overhead.

 [1] pFedMMA: Personalized Federated Fine-Tuning with Multi-Modal Adapter for Vision-Language Models, ICLR 2026

**Strengths And Weaknesses:**

Strengths:
1) Clear main thread: The paper first proposes that existing works on LLM fine-tuning in FL with LoRA mainly focus on horizontal heterogeneity (Non-IID), neglecting functional heterogeneity across LLM layers. Then, the paper thoroughly investigates the dual-heterogeneity from motivational studies, method design, and experimental validation, demonstrating that the proposed tree-structured aggregation can reconcile statistical and functional heterogeneity.
2) Interesting method innovation: The alignment between the layer-wise hierarchy of LLMs and aggregation granularity is interesting and appealing. By studying the optimal degree of parameter sharing, which varies with model depth, and sharing shallow generic features and personalizing deep semantic reasoning, the method provides a more expressive tree-structured aggregation than previous strategies. The cluster-external expert design further enables efficient personalization and generalization.
3) Well-written and plentiful experiments: The paper vividly describes shallow features and deep features as the root and branches of a hierarchical tree, naturally integrating the tree-structured framework with LLM layers. Then, it compares the client similarity measurement across all LLM layers to a global topology, further connecting it to the tree structure. Moreover, theoretical convergence analysis and comprehensive experiments (including extensive experiments in the appendix) further demonstrate the effectiveness of the proposed framework.

Weaknesses:
1) Ambiguity between motivation and method design: It is unclear whether the functional heterogeneity should be considered a heterogeneity challenge comparable to statistical heterogeneity (for example, model heterogeneity is a proper vertical heterogeneity challenge comparable to statistical heterogeneity). The functional heterogeneity is more like a change in the internal model structure caused by the dynamic variation of the cluster number at each layer in the tree structure during method design, rather than an essential heterogeneous challenge in motivation. If it is a new heterogeneity challenge, it remains unclear how baselines behave under this challenge.
2) High computational overhead: The method requires pairwise client similarity computation, which grows rapidly as the client number increases and further causes a large amount of computational overhead. Moreover, the topological structure is constructed only once after warmup. If the degree of heterogeneity is high (the paper set the parameter of Dirichlet from 0.5 to 1, which is relatively low statistical heterogeneity), the early personalized training tends to overfit, which may cause the warmup to fail and thereby affect the model performance.
3) Citation format issues: The citation format of the paper can be improved. For example, plural nouns are used when citing only a single reference (e.g., right half of L25–26 and L87–88). Although multiple references are cited later, the grammatical number of the noun should match the number of sources at the current citation point. Additionally, multiple references are listed in L34–36 (right half), they should be arranged in chronological order.

---

> ### Author Rebuttal · Authors · 2026-03-29
>
> # **Response to Reviewer TqYS**
>
> We thank the reviewer for the detailed and insightful comments. We address each point below.
>
> ---
>
> ### 1. Ambiguity between motivation and method design
>
> We appreciate this clarification. Our intention is not to introduce functional heterogeneity as a new standalone heterogeneity type comparable to statistical or model heterogeneity. Instead, it describes an emergent phenomenon in Transformer-based fine-tuning: even with identical architectures, different layers exhibit different sensitivity to client heterogeneity.
>
> Our contribution is to show that horizontal heterogeneity interacts with this layer-wise functional variation, leading to depth-dependent aggregation behavior (Fig. 2). The method is then designed to address this interaction, rather than redefining the heterogeneity taxonomy.
>
> ---
>
> ### 2. Computational overhead and warm-up reliability
>
> We clarify that the tree construction is performed **only once after warm-up**, and no further pairwise similarity computation is required during training, keeping the overhead limited.
>
> Regarding heterogeneity, Dirichlet $\alpha \in [0.5,1]$ already induces substantial label skew for GLUE tasks (binary/ternary classification), yielding meaningful non-IID settings, consistent with prior work such as FedSA (ICLR 2025). We also evaluate more challenging **task-level heterogeneity** (Appendix, p.21), where clients are assigned different tasks, increasing divergence. Our method remains stable under these settings.
>
> Concerning warm-up reliability, our empirical results (**Response to Reviewer RTT1 (Q2)** ) show that a short warm-up is sufficient to recover a stable structure, and extending warm-up does not significantly change the performance.
>
> ---
>
> ### 3 & 4. Citation format and clarification on clustering works
>
> We thank the reviewer for pointing out these issues. We will carefully revise the citation format in the final version, including grammatical consistency and chronological ordering.
>
> Regarding the specific point, we clarify that only FedLEASE explicitly adopts client clustering. We will revise the wording to avoid ambiguity.
>
> ---
>
> ### 5. Reconciling generalization and personalization
>
> The reconciliation refers to achieving both simultaneously within the same model, rather than a trade-off across tasks.
>
> This is most clearly demonstrated in the NLG setting, where we evaluate:
> - performance on each task data (personalization), and
> - overall/generalized performance (average across clients).
>
> As shown in Table 2, our method improves both local performance and harmonic mean (which jointly measures generalization and personalization), indicating that the design does not sacrifice one for the other, but balances them within a unified framework.
>
> ---
>
> ### 6. Importance of tree-structured alignment
>
> The importance of the tree structure is supported by multiple ablations.
>
> In addition to Table 3, Table 4 shows that **fixed-depth sharing is suboptimal**, indicating that aggregation should vary across layers. Table 5 further shows that removing structural consistency (i.e., breaking the hierarchical design) leads to performance degradation.
>
> Together, these results demonstrate that the tree structure is not only beneficial but necessary to capture the depth-dependent aggregation pattern.
>
> ---
>
> ### 7. Applicability to multimodal settings (pFedMMA)
>
> We thank the reviewer for pointing out this relevant work. Interestingly, pFedMMA makes observations that are **consistent with our findings**.
>
> Specifically, pFedMMA notes that *higher layers contain more discriminative and dataset-specific features, while lower layers preserve general knowledge*. This aligns with our observation that deeper layers are more sensitive to heterogeneity.
>
> Moreover, their method does not fully share deep layers. Instead, only the **shared projection layer is aggregated**, while the up- and down-projection layers remain personalized. This effectively corresponds to **partial sharing at higher layers**, rather than full deep-layer aggregation.
>
> Therefore, rather than contradicting our findings, pFedMMA supports the same principle: deeper layers require more careful, often partial or structured sharing.
>
> ---
> ### 8. Handling client dynamics
>
> We agree that dynamic participation is an important practical scenario. This setting is closer to federated continual learning and is not the primary focus of our work.
>
> We refer the reviewer to **Response to Reviewer RTT1 (Q3)** for a detailed discussion, including both the extension mechanism and experimental results. In brief, a new client can be integrated via short warm-up, similarity-based branch matching, and subsequent layer-wise aggregation. Client departure is simpler, as it only removes the client from aggregation without requiring changes to the tree structure.
>
> The results show that new clients can be incorporated with negligible impact on existing clients, while achieving substantial gains over local-only training.

---

> > ### Author Rebuttal · Reviewer_TqYS · 2026-04-01
> >
> > The authors’ responses have adequately addressed my concerns on motivation, references, computation overhead, personalization & generalization, and ablation study. Therefore, I will retain my "Weak Accept" recommendation.

---

> > > ### Author Response · Authors · 2026-04-01
> > >
> > > We sincerely thank the reviewer for the positive feedback and for acknowledging that our responses have addressed the concerns. We are glad that our clarifications were helpful. We appreciate the reviewer’s time and thoughtful evaluation.

---

### Official Review · Reviewer_UYeW · 2026-03-07

**Soundness:** 3
**Presentation:** 3
**Significance:** 3
**Originality:** 3
**Overall Recommendation:** 5
**Confidence:** 4

**Summary:**

This paper proposes a personalized federated fine-tuning framework that adapts to non-iid data distribution. The author first points out two key observations in federated aggregation under client data heterogeneity. First, aggregation over different layer sets heavily affects the final accuracy due to the distinct roles of different layers (shallow and deep). Second, the effect of different layer-aggregation is determined by the underlying data distribution, and the optimal set of layers can shift entirely from deep to shallow. Motivated by these observations, the author proposes FedTreeLoRA, a tree-based approach that captures general knowledge at the root and task-specific knowledge at the branches. The optimal per-layer cluster is identified via a search space dependent on the layer index, such that shallow layers cluster closer to the root and deeper layers cluster further from the root. The per-layer aggregation strategy is designed as a weighted sum of the internal and external LoRA experts. The paper provides convergence analysis and demonstrates the effectiveness of FedTreeLoRA through extensive experiments on NLU and NLG tasks.

**Compliance With Llm Reviewing Policy:**

Affirmed.

**Final Justification:**

The author have addressed my questions, and I maintain my score.

**Key Questions For Authors:**

1. Does the motivational observation generalize to other models? It would be helpful if the author can conduct the same experiments from Figures 1 and 2 on more modern LLMs to verify the generalization of the trend.
2. Why does FedTreeLoRA have the same percentage of trainable parameters as FedIT in Tables 1 and 2? The proposed method additionally learns a dynamic mixing coefficient per layer, which supposes to add some extra trainable parameters compared to a naive federated LoRA training.
3. How does FedTreeLoRA adapt to client sampling in federated learning?

**Limitations:**

yes

**Strengths And Weaknesses:**

Strengths:
* The paper is clearly written with detailed discussion on prior literature and the missing gap.
* Motivational studies are thoroughly discussed and demonstrated with example experiments. The observations made are indeed fascinating and insightful. This could be a valuable empirical contribution that opens for more future research ideas on personalized learning.
* The methodology of FedTreeLoRA are well described and easy to follow and understand.
* Experimental study is executed in depth and the results look promising. The author also provides extensive ablation study.

Weaknesses:

* The proposed method requires some hyper-parameter tuning, including window-size $K$ and heterogeneity threshold $\tau$. This may adds overhead and reduces the ease of use across new LLMs and datasets.
* Experiments do not consider client subsampling which is common in federated learning.
* The paper provides insightful findings on the duel heterogeneity and proposes a method to address the observed problems accordingly. However, I personally find the proposed method itself less significant than the motivational studies, and the accuracy improvement is modest (within 1-2% compared to the second best method).

---

> ### Author Rebuttal · Authors · 2026-03-29
>
> # **Response to Reviewer UYeW**
>
> We thank the reviewer for the constructive feedback and helpful suggestions.
>
> ### 1. Hyperparameter sensitivity (τ and K)
>
> We acknowledge that the method introduces two hyperparameters (τ and K). However, as shown in Appendix C.1, FedTreeLoRA exhibits a **robustness plateau**: performance remains stable within a wide range (e.g., $\tau \in [0, 0.06]$ and $K \in \{3,4,5\}$), and consistently outperforms baselines even under suboptimal settings. In practice, we use fixed values across all GLUE experiments without task-specific tuning, which keeps the method easy to apply.
>
> ---
>
> ### 2. Missing client subsampling experiments
>
> We add an experiment with client subsampling on the GLUE benchmark.
>
> **Setup.** We simulate a system with 20 clients, where at each communication round, 5 clients are randomly sampled to participate. All other settings follow the main NLU setup.
>
> **Results:**
>
> | Method | MNLI | QNLI | SST2 | QQP | Average |
> |---|---:|---:|---:|---:|---:|
> | FedIT | 82.25 | 86.95 | 93.39 | 83.82 | 86.60 |
> | FFA-LoRA | 82.50 | 87.97 | 94.20 | 83.97 | 87.16 |
> | FedSA | 83.85 | 90.35 | 95.72 | 89.05 | 89.74 |
> | FedDPA | 84.13 | 90.78 | 94.52 | 89.10 | 89.63 |
> | FedALT  | 84.77 | 89.28 | 95.54 | 88.98 | 89.64 |
> | FedLEASE | 85.28 | 91.83 | 94.68 | 89.75 | 90.39 |
> | FedTreeLoRA | 87.10 | 92.98 | 95.93 | 90.30 | 91.58 |
>
> The results show that FedTreeLoRA maintains consistent improvements under partial participation, demonstrating its robustness to client subsampling.
>
> ---
>
> ### 3. Significance of method vs. motivational findings
>
> We appreciate this perspective. While the motivational studies are important, they directly guide the design of our method. The improvement over strong baselines is not only consistent but achieved with **negligible additional parameters and minimal overhead**.
>
> More importantly, the contribution lies in introducing a new perspective: **aggregation should be jointly determined by data heterogeneity and model depth**. Our method provides a concrete mechanism to realize this, and the consistent gains across both NLU and NLG tasks suggest that the design is effective rather than incremental.
>
> ---
>
> ### 4. Generalization of motivational observations
>
> In addition to the original motivational experiments on GLUE using RoBERTa, we here extend the analysis to LLaMA-2-7B on FLAN tasks. For Motivation 1, we consider two tasks, Natural Language Inference (NLI) and Sentiment Analysis (SA). Each task involves four clients, where two datasets are assigned within the same task (for example, SNLI and QNLI for NLI) to simulate a non-IID setting.
>
> For Motivation 2, we define three levels of heterogeneity. In the homogeneous setting, all four clients use the same dataset within the NLI task. In the moderate setting, clients are split across two different datasets within the NLI task. In the heterogeneous setting, clients are assigned datasets from both NLI and SA tasks.
>
> **Results:**
>
> *Motivation 1: shallow vs. deep aggregation*
>
> | Setting | Local | Shallow | Deep | Full |
> |---|:---:|:---:|:---:|:---:|
> | NLI | 67.69 | **70.72** | 67.22 | 69.03 |
> | SA | 61.08 | **62.20** | 59.88 | 61.72 |
>
> *Motivation 2: effect of heterogeneity on optimal depth*
>
> | Setting | 8 layers | 16 layers | Full |
> |---|:---:|:---:|:---:|
> | Homogeneous | 67.48 | 68.63 | **69.07** |
> | Moderate | 67.11 | **69.71** | 66.47 |
> | Heterogeneous | **61.27** | 59.42 | 57.22 |
>
> The results follow the same trend as in Figure 1 and Figure 2, indicating that the observations generalize beyond RoBERTa to modern LLMs.
>
> ---
>
> ### 5. Trainable parameter count
>
> We thank the reviewer for pointing this out. The additional parameters introduced by the dynamic mixing coefficients are extremely small.
>
> Specifically, the mixing coefficient is a scalar per layer and per client, resulting in only a **~0.02% increase relative to LoRA parameters** (as shown in Table 3). Since LoRA itself occupies only a small fraction of the full model (e.g., 0.1107% in Table 1), this additional overhead is negligible (0.1107% * 0.02%) and does not affect the reported percentage when rounded to four decimal places.

---

> > ### Author Rebuttal · Reviewer_UYeW · 2026-04-01
> >
> > Thank you for conducting the experiment in such a short period! I have no further question and I will maintain my positive score.

---

> > > ### Author Response · Authors · 2026-04-01
> > >
> > > We sincerely thank the reviewer for the positive feedback. We are glad that our responses have addressed the concerns. We appreciate the reviewer’s time and thoughtful evaluation.

---

### Official Review · Reviewer_XhGi · 2026-03-08

**Soundness:** 3
**Presentation:** 3
**Significance:** 2
**Originality:** 3
**Overall Recommendation:** 4
**Confidence:** 4

**Summary:**

This paper introduces FedTreeLoRA, a personalized federated fine-tuning framework that addresses the interplay between statistical data heterogeneity across clients and the inherent functional heterogeneity of LLMs. Challenging the Flat-Model Assumption that treats models monolithically, the authors empirically show that optimal parameter sharing is depth-dependent. To leverage this, FedTreeLoRA builds a global client hierarchy from LoRA B-matrices and employs an adaptive, layer-wise clustering strategy. This creates a tree structure where layers are specialized for client subgroups. The method achieves state-of-the-art results on NLU and NLG benchmarks with minimal overhead, supported by theoretical convergence guarantees.

**Compliance With Llm Reviewing Policy:**

Affirmed.

**Final Justification:**

The authors’ response has adequately addressed my concerns.

**Key Questions For Authors:**

1. What is the marginal performance gain of FedTreeLoRA's complex tree structure compared to a simpler, optimized "Layer-wise FedPer" baseline (i.e., using hyperparameter search to find a single optimal split layer $L_{split}$ for all clients)? Is the added complexity of building a full tree justified by the performance improvement?

2. In line 197, the authors claim that “the hierarchical prior that specialization increases with depth”, which lacks citation or experiment to verify. In Figure 8, it can be seen that Lora B actually doesn’t exhibit a simple linear relationship with Transformer shallow layers in the NonIID setting, and the authors use it to judge specialization. Is this in conflict with the statement above? Further, the constraint $c_{l-1}^* \le c_{l}^*$ may sometimes be suboptimal?

3. Can the authors prove that after the warmup phase, the tree structure is almost the optimal structure? Will adding more criterion,e.g., lora A, to judge the similarity and form the tree structure be better?

**Limitations:**

Yes

**Strengths And Weaknesses:**

Strengths:

1. The paper identifies both horizontal heterogeneity and vertical heterogeneity, highlighting the coupling relationship between the two. It argues that the optimal depth of parameter sharing should depend on the degree of data similarity. Motivational studies in Section 4 further support the insight. These elements integrate the proposed method into an organic whole.

2. The proposed method is insightful, especially the tree structure. It forms a progressive clustering structure layer by layer, meanwhile avoiding that client groupings fluctuate chaotically between adjacent layers.

3. The experiment shows that the proposed method achieves SOTA performance, and each component is proven to be effective by ablation studies. More importantly, employing the tree structure alone can even get SOTA performance.

4. The paper proves an $O(1/\sqrt{T})$ convergence rate under standard smooth non-convex assumptions.

Weaknesses:

1. Horizontal heterogeneity is a problem coming from federated learning, and vertical heterogeneity is an intrinsic property of LLMs. While the paper successfully bridges the relationship between horizontal and heterogeneity heterogeneity, neither dimension is novel in isolation, not a new problem raised when combining FL and PEFT, which weakens the contribution of this work.

2. The core idea to solve vertical heterogeneity, sharing shallow layers, is proposed in previous work like FedPer. Though the author claim that “Transformer-based LLMs consist of architecturally identical layers where hierarchy emerges from semantic specialization, which is diffetent from CNNs used in previous work”. I have find special modifications for this architecture. Further the author claims that they solve “how LoRA parameters should be aggregated differentially across Transformer depth based on client similarity“, but using the LoRA B matrix as a proxy for task similarity is a common practice.

3. The global tree structure is static, constructed once after the warmup phase. As client similarities can drift during training, relying solely on initial B matrix updates to determine the topology for the entire training process may lead to suboptimal clustering in later stages.

4. More experiements should be added, e.g., the learnable parameter $\lambda_{l,k}$ is suggested to be reported to understand how the model balances cluster-specific and external knowledge.

5. The framework seems to be overly complex and introduces specific hyperparameters such as the search range $K$ and the heterogeneity threshold $\tau$. The algorithm's reliance on these thresholds makes it time-consuming to implement compared to simpler baselines.

---

> ### Author Rebuttal · Authors · 2026-03-29
>
> # **Response to Reviewer XhGi**
>
> We thank the reviewer for the detailed and constructive feedback. Below we address each concern.
>
> ### 1. Novelty of dual heterogeneity
>
> We agree that horizontal (data) and vertical (model depth) heterogeneity are individually known. Our contribution is to identify their coupled interaction: the optimal aggregation depth depends on client similarity (Fig. 2), implying that aggregation should be jointly determined by data heterogeneity and model depth.
>
> We therefore introduce a **tree-structured aggregation mechanism** that adaptively determines layer-wise sharing. To our knowledge, this coupling and its algorithmic realization have not been explored in prior FL+PEFT work.
>
> ---
>
> ### 2. Relation to FedPer and use of LoRA B
>
> FedPer assumes a *single global split*, which cannot adapt to varying client similarity. In contrast, our method determines aggregation depth dynamically and allows different client groups to diverge at different layers.
>
> While LoRA B has been used in prior work, our contribution lies in using it to construct a globally consistent hierarchical topology for layer-wise aggregation.
>
> ---
>
> ### 3. Static tree structure
>
> We refer the reviewer to **Response to Reviewer RTT1 (Q2)** for detailed justification and empirical validation. In brief, the warm-up phase already provides a stable and semantically meaningful estimate of client relationships, and extending the warm-up does not significantly change performance.
>
> Regarding the similarity metric, we use LoRA B based on both empirical evidence and efficiency considerations. Prior work (FedLease, NeurIPS 2025) shows that B captures task-specific variations, while A reflects more general features. Although BA could be used, it requires full-rank reconstruction and incurs higher cost, making B a practical choice for federated fine-tuning.
>
> ---
>
> ### 4. Analysis of learnable parameter λ
>
> We thank the reviewer for this suggestion. Table 3 shows that even the isolationist variant outperforms strong baselines, indicating that the primary gain comes from accurate tree-structured grouping. We further analyze the learned routing coefficient by reporting the average cluster weight across shallow, middle, and deep layers on SST2. Due to space limitations, we report three example clients (query-side) here.
>
> | Client | Shallow (0–7) | Middle (8–15) | Deep (16–23) |
> |---|---:|---:|---:|
> | Client 1 | 1.0000 | 0.7409 | 0.7934 |
> | Client 2 | 1.0000 | 0.7383 | 0.8630 |
> | Client 3 | 1.0000 | 0.7261 | 0.9229 |
>
>
> In shallow layers, the cluster count is 1 (see Figure 4), so all clients share the same root expert and the corresponding weight is naturally 1. After branching begins, clients place substantially higher weight on their own cluster expert, showing that the model prefers cluster-specific knowledge once heterogeneity becomes more relevant. This trend is especially clear in the deep layers, which supports our claim that deeper representations require more selective sharing.
>
> ---
>
> ### 5. Complexity and hyperparameters
>
> We refer the reviewer to **Response to Reviewer UYeW (Q1)**.
>
> ---
>
> ### 6. Comparison to optimized Layer-wise FedPer
>
> FedPer-style methods rely on selecting a *single global split layer* k as a tunable hyperparameter. To provide a strong baseline, we perform a grid search over k ∈ [4, 20] and report the *best-performing split per dataset* on GLUE:
>
> | Method | MNLI | QNLI | SST2 | QQP |
> |---|:---:|:---:|:---:|:---:|
> | FedPer (best k) | 83.60 (k=7) | 89.65 (k=12) | 95.04 (k=20) | 88.70 (k=8) |
> | Ours | 88.15 | 93.37 | 96.56 | 91.35 |
>
> This highlights two key limitations:
>
> (1) Dataset-specific tuning: the optimal k varies significantly across tasks, indicating no universal split.
>
> (2) Structural limitation: even with oracle tuning, FedPer enforces a *flat split* (shared before k, personalized after k), which cannot capture the coupling between statistical and functional heterogeneity.
>
> In contrast, FedTreeLoRA constructs a global tree and determines layer-wise aggregation along that topology, producing a hierarchical sharing pattern from shared shallow trunks to specialized deep branches.
>
> ---
> ### 7. Monotonicity constraint and specialization claim
>
> Our claim that specialization increases with depth refers to a structural tendency at the aggregation level, not strict monotonic similarity.
>
> Fig. 2 shows that as heterogeneity increases, the optimal sharing boundary shifts toward shallower layers, indicating that deeper layers are more sensitive to heterogeneity.
>
> The monotonicity constraint is applied to the **cluster count \(c_l\)**, enforcing a consistent transition from shared to specialized representations. Without this constraint, clusters may reshuffle across layers, causing instability. As shown in Table 5, removing structural consistency degrades performance.
>
> Therefore, the constraint ensures a stable and coherent hierarchical aggregation structure rather than imposing an unrealistic assumption.

---

> > ### Author Rebuttal · Reviewer_XhGi · 2026-04-02
> >
> > The authors’ response has adequately addressed my concerns, and I will raise my score.

---

> > > ### Author Response · Authors · 2026-04-02
> > >
> > > We sincerely thank the reviewer for the positive feedback and for raising the score. We appreciate the reviewer’s time and consideration.

---

### Official Review · Reviewer_RTT1 · 2026-03-13

**Soundness:** 3
**Presentation:** 3
**Significance:** 3
**Originality:** 3
**Overall Recommendation:** 4
**Confidence:** 4

**Summary:**

The paper proposes FedTreeLoRA, a tree-structured aggregation framework for federated fine-tuning that aligns aggregation granularity with the layer hierarchy of LLMs to handle statistical and functional heterogeneity while improving performance with minimal parameter overhead.

**Compliance With Llm Reviewing Policy:**

Affirmed.

**Final Justification:**

The authors’ response has adequately addressed my concerns, and I will retain my positive score.

**Key Questions For Authors:**

1.While the empirical results in Figure 2 suggest a tendency to aggregate shallow layers, could occasionally aggregating deeper layers benefit performance for certain tasks? Does the proposed tree structure account for such cases?

2.Is the global distance or tree structure updated after the warm-up stage? If not, what is the rationale behind keeping it fixed?

3.How does the system support clients who join or participate during the middle of training?

4.Some related works on gradient-based layer-wise selection and aggregation appears to be missing, such as Fed-HeLLo [1]. Please discuss this line of work and clarify the differences between gradient-based and similarity-based layer modeling.

[1] Fed-HeLLo: Efficient Federated Foundation Model Fine-Tuning with Heterogeneous LoRA Allocation

**Limitations:**

More diverse LLM architectures could be considered in the evaluation to further demonstrate the generality of the proposed tree-structured aggregation method.

**Strengths And Weaknesses:**

Strengths:

1.The paper identifies and formulates the important problem of dual heterogeneity in federated LLM fine-tuning, highlighting the interaction between statistical heterogeneity across clients and functional heterogeneity across model layers.

2.The proposed FedTreeLoRA introduces a novel tree-structured aggregation mechanism that adaptively determines the optimal parameter sharing depth across layers, effectively balancing global knowledge sharing and client-level personalization.

3.The method demonstrates consistent empirical improvements across multiple NLU and NLG benchmarks while maintaining negligible parameter overhead, showing practical efficiency for scalable federated fine-tuning.

Weakness:

See Questions

---

> ### Author Rebuttal · Authors · 2026-03-29
>
> # **Response to Reviewer RTT1**
>
> We thank the reviewer for the insightful questions and address them below.
>
> ---
>
> ### 1. When is deeper-layer aggregation beneficial? Does the tree capture this?
>
> We agree that deeper-layer aggregation is beneficial when client distributions are highly similar, this is precisely one of our motivations.
>
> Our empirical findings (Fig. 1–2) show that the *optimal sharing depth is not fixed*, but **coupled with client similarity**:
> - Under homogeneous settings, aggregating deeper layers improves performance;
> - As heterogeneity increases, the optimal boundary shifts toward shallower layers.
>
> FedTreeLoRA explicitly models this coupling via **layer-wise adaptive aggregation**. Instead of enforcing a fixed sharing rule, it determines the aggregation depth per layer based on client similarity. As a result, similar clients remain grouped deeper, while dissimilar clients diverge earlier.
>
> Importantly, we enforce a **globally consistent tree structure** with a monotonic constraint. Without this, arbitrary regrouping across layers can lead to *topological inconsistency* (e.g., cluster reshuffling across depth), disrupting semantic continuity and causing unstable updates. Our design ensures a coherent progression from shared shallow “trunks” to specialized deep “branches.”
>
> ---
>
> ### 2. Is the tree updated after warm-up? Why keep it fixed?
>
> In our design, the tree is constructed after warm-up and then kept fixed.
>
> The key reason is that warm-up already provides a **meaningful estimate of client relationships**, rather than a rough initialization. As shown in Appendix (Fig. 7), the learned hierarchy aligns well with task semantics: clients share a single cluster in shallow layers, then progressively split into task-related groups at deeper layers. This indicates that the tree captures a stable and interpretable structure.
>
> We further validate that a short warm-up is sufficient:
>
> | Warm-up | MNLI | QNLI | QQP | SST-2 | Avg |
> |---|---:|---:|---:|---:|---:|
> | 5 | 82.94 | 89.31 | 84.75 | 94.19 | 87.80 |
> | 10 | 82.66 | 89.83 | 84.13 | 94.62 | 87.81 |
>
> Performance remains nearly unchanged, suggesting early topology discovery is sufficient.
>
> Based on this, we decouple **topology discovery (warm-up)** from **parameter optimization (training)**. Keeping the tree fixed preserves structural consistency and avoids repeated similarity computation and clustering, reducing overhead.
>
> ---
>
> ### 3. How to handle clients joining mid-training?
>
> We agree this is an important practical scenario, though closer to federated continual learning and not the primary focus of this work.
>
> Nevertheless, FedTreeLoRA can be naturally extended. A new client can:
> (1) perform a short local warm-up,
> (2) compute its LoRA \(B\) representation,
> (3) match to the closest branch in the existing tree based on similarity,
> (4) follow the same layer-wise aggregation thereafter.
>
> We include an experiment on MNLI where a new client joins at round 10 or 20. For each setting, we randomly sample a different client to join the system, to avoid bias from a specific client. We report: (i) the performance of original clients without the new client, (ii) the performance of existing clients after the new client joins, (iii) the performance of the new client after joining, and (iv) a local-only baseline for the new client.
>
> | Join round | Original clients (no join) | Existing clients (after join) | New client (after join) | New client (local only) |
> |---|---:|---:|---:|---:|
> | 10 | 87.90 | 87.43 | 84.50 | 82.00 |
> | 20 | 87.90 | 87.50 | 86.50 | 79.00 |
>
> The results show negligible impact on existing clients and significant improvement over local-only training, indicating that the learned tree provides a robust reusable topology.
>
> ---
>
> ### 4. Relation to Fed-HeLLo
>
> Thanks for pointing out this relevant work. We will include it in the revision.
>
> The key difference is that Fed-HeLLo and our method address different problems.
>
> Fed-HeLLo focuses on **resource heterogeneity**, deciding which layers each client should train based on gradient/FIM importance under limited budgets. Its goal is layer allocation under resource constraints, while aggregation remains standard layer-wise averaging.
>
> In contrast, FedTreeLoRA focuses on **statistical and functional heterogeneity across clients** and addresses a different question: *which clients should share parameters at each layer*. We construct a similarity-based global tree and perform **hierarchical, layer-wise aggregation**, which determines the optimal sharing boundary across depth.
>
> In summary:
> - **Fed-HeLLo**: gradient-based layer importance → resource-aware allocation
> - **FedTreeLoRA (ours)**: similarity-based client relations → hierarchical aggregation topology
>
> Fed-HeLLo does not model hierarchical client structure nor depth-dependent sharing boundaries, which are central to our approach.

---

> > ### Author Rebuttal · Reviewer_RTT1 · 2026-03-31
> >
> > The authors’ response has adequately addressed my concerns, and I will retain my positive score.

---

> > > ### Author Response · Authors · 2026-03-31
> > >
> > > We sincerely thank the reviewer for the positive feedback and for acknowledging that our responses have addressed the concerns. We also appreciate the constructive discussion and the reviewer’s time and consideration.

---

### Decision · Program_Chairs · 2026-04-30

**Decision:**

Accept (regular)

**Comment:**

The paper received uniformly positive reviews, with one accept and three weak accepts, and the reviewers’ concerns were largely resolved after rebuttal. This paper proposes FedTreeLoRA, a personalized federated fine-tuning framework that models the interaction between statistical heterogeneity across clients and depth-wise functional heterogeneity in LLMs through a tree-structured, layer-wise aggregation strategy. Reviewers appreciated the clear motivation, the insightful formulation of depth-dependent parameter sharing, the well-designed tree structure for balancing shared and personalized knowledge, and the strong empirical results across both NLU and NLG tasks with minimal parameter overhead. The paper was also viewed as technically solid, with supportive ablations, motivating studies, and useful theoretical analysis.
The main concerns focused on limited novelty relative to existing personalized FL and layer-sharing ideas, the added complexity and hyperparameter tuning required by the framework, and questions about the use of a static tree built after warm-up, scalability under client dynamics or subsampling, and generality to broader architectures or multimodal settings. Some reviewers also asked for stronger comparisons to simpler layer-wise baselines and clearer positioning with respect to related work. However, the rebuttal was generally considered effective and addressed these concerns to the reviewers’ satisfaction, with no reviewer lowering their score and one reviewer explicitly raising it. Overall, the paper makes a meaningful and well-supported contribution to federated LLM fine-tuning, and the AC recommends accepting the paper.